# Zebrafish Posterior Lateral Line primordium migration requires interactions between a superficial sheath of motile cells and the skin

Damian E Dalle Nogare[1]*, Naveen Natesh[1†], Harshad D Vishwasrao[2], Hari Shroff[2,3], Ajay B Chitnis[1]

[1]Section on Neural Developmental Dynamics, Division of Developmental Biology, Eunice Kennedy Shriver National Institute of Child Health and Human Development, National Institutes of Health, Bethesda, United States; [2]Advanced Imaging and Microscopy Resource, National Institutes of Health, Bethesda, United States; [3]Laboratory of High Resolution Optical Imaging, National Institute of Biomedical Imaging and Bioengineering, National Institutes of Health, Bethesda, United States

**Abstract** The Zebrafish Posterior Lateral Line primordium migrates in a channel between the skin and somites. Its migration depends on the coordinated movement of its mesenchymal-like leading cells and trailing cells, which form epithelial rosettes, or protoneuromasts. We describe a superficial population of flat primordium cells that wrap around deeper epithelialized cells and extend polarized lamellipodia to migrate apposed to the overlying skin. Polarization of lamellipodia extended by both superficial and deeper protoneuromast-forming cells depends on Fgf signaling. Removal of the overlying skin has similar effects on superficial and deep cells: lamellipodia are lost, blebs appear instead, and collective migration fails. When skinned embryos are embedded in Matrigel, basal and superficial lamellipodia are recovered; however, only the directionality of basal protrusions is recovered, and migration is not rescued. These observations support a key role played by superficial primordium cells and the skin in directed migration of the Posterior Lateral Line primordium.

*For correspondence: damian.dallenogare@nih.gov

Present address: †Center for Genomics and Computational Biology and Department of Biomedical Engineering, Duke University, Durham, United States

Competing interests: The authors declare that no competing interests exist.

Reviewing editor: Holger Knaut,

## Introduction

Collective cell migration is a fundamental process for embryonic development (*Szabó and Mayor, 2018*) and its dysregulation during morphogenesis is a key contributor to many developmental disorders (*Friedl and Gilmour, 2009*). Recent studies have additionally placed collective cell migration at the heart of metastasis of certain types of cancer (*Clark and Vignjevic, 2015*; *Wang et al., 2016*). These physiological and pathological contexts are linked by the necessity for cells to migrate using unique strategies as they navigate through diverse 3D environments, while maintaining group cohesion and directionality.

In recent years, the Zebrafish Posterior Lateral Line primordium (PLLp) has emerged as a powerful model for studying a wide range of cellular and developmental processes, including cell-cell signaling, tissue patterning, and collective migration (*Chitnis et al., 2012*; *Friedl and Gilmour, 2009*; *Ghysen and Dambly-Chaudière, 2007*). This group of 100–150 cells is initially specified adjacent to the otic vesicle and migrates caudally down the length of the embryo over the course of ~24 hr (*Gompel et al., 2001*; *Nogare et al., 2017*). As this primordium migrates, cells in the trailing domain are progressively reorganized into apically constricted epithelial rosettes, each cradling a central sensory hair cell progenitor (*Nechiporuk and Raible, 2008*). The trailing-most cells in the PLLp

eventually lose the ability to sustain collective migration and are deposited by the migrating primordium. Cells that had been incorporated into epithelial rosettes are deposited as neuromasts and will go on to develop into the mature sense organs of the lateral line, a sensory system that detects water flow about the animal (*Coombs and Van Netten, 2005*). Cells in the PLLp that are not incorporated into these epithelial rosettes are deposited as a continuous stream of so-called 'interneuromast cells', which lie between the sense organs (*Grant et al., 2005*; *López-Schier and Hudspeth, 2005*; *Nogare et al., 2017*).

Migration of the PLLp is guided by a stripe of chemokine, Cxcl12a, secreted by muscle pioneer cells lying along the horizontal myoseptum (*David et al., 2002*; *Li et al., 2004*). Directional interpretation of this uniform stripe is accomplished by the expression of two chemokine receptors in the PLLp, Cxcr4b, and Ackr3b (previously Cxcr7b). Cxcr4b binds Cxcl12a and engages the G-protein-coupled signaling transduction pathway to determine directed migration in response to a Cxcl12a. Expression of Ackr3b, an atypical chemokine receptor which can bind Cxcl12a with high affinity but cannot activate G-protein-coupled signaling to mediate migration is restricted to trailing cells in the primordium (*Boldajipour et al., 2008*; *Burns et al., 2006*; *Naumann et al., 2010*). It internalizes and degrades bound Cxcl12a generating a local gradient of Cxcl12a with lower levels toward the trailing end and higher levels toward the leading end of the primordium (*Dambly-Chaudière et al., 2007*; *David et al., 2002*; *Haas and Gilmour, 2006*; *Lewellis et al., 2013*; *Wong et al., 2020*). Although transcripts for *cxcr4b* are restricted to the leading two-thirds of the PLLp, the receptor itself is broadly expressed along the entire length of the primordium and it gives primordium cells the potential for directed migration in response to the self-generated Cxcl12a gradient (*Donà et al., 2013*; *Venkiteswaran et al., 2013*).

In cell transplantation experiments, basal cryptic lamellipodia are observed extending from PLLp cells in the direction of migration (*Haas and Gilmour, 2006*; *Lecaudey et al., 2008*), a common strategy for migrating epithelial cells (*Farooqui and Fenteany, 2005*). Crucially, these lamellipodia are observed extending from both leading cells, which have a more mesenchymal morphology, and from the basal feet of epithelial cells, which have a more typically epithelial morphology (*Haas and Gilmour, 2006*), suggesting that cells along the length of the PLLp actively contribute to migration. This is consistent with recent studies showing that chemokine signaling is necessary along the entire Cxcr4b-expressing domain to support effective collective migration (*Colak-Champollion et al., 2019*). In addition to chemokine signaling, Fibroblast growth factor (Fgf) signaling is also required for migration. The polarization of these basal migratory protrusions appears to be dependent on Fgf signaling in response to Fgfs produced in the leading part of the primordium. Their polarity is lost upon Fgf receptor inhibition, even when chemokine signaling is unperturbed, and this occurs concomitantly with a loss of migratory ability (*Lecaudey et al., 2008*). Furthermore, experiments with isolated PLLp fragments generated by laser ablation suggest that Fgf could act as a direct migratory cue (*Dalle Nogare et al., 2014*). These two systems, and potentially others, act together to govern collective migration of the PLLp.

Apart from the fact that underlying muscle pioneer cells are the source of chemokine signals that guide the primordium, the manner in which the PLLp interacts with surrounding tissue as it migrates and what influence surrounding tissue might have on migration and morphogenesis remains poorly understood. Aman et al showed that traversing underlying intersomitic boundaries does not influence the deposition of neuromasts, as the lateral line primordium does not deposit more closely spaced neuromasts in *trilobyte* mutants, which have more densely packed somites (*Aman et al., 2011*). Other studies have shown that the directionality of primordium migration does not rely on any extrinsic cues from the surrounding tissue and that its directional migration is an autonomous property of the primordium itself (*Haas and Gilmour, 2006*). However, the primordium has a dramatic effect on the tissue through which it migrates. The PLLp migrates along the horizontal myoseptum, between the underlying somites and overlying skin. As it migrates, the skin is displaced upwards and is separated from the underlying tissue by the passage of the PLLp, returning rapidly to its original apposition with the underlying somites after the passage of the PLLp.

In this study, we focus on flat superficial PLLp cells that lie above the deeper epithelialized cells that form protoneuromasts. We show that these cells extend directional migratory processes apposed to the overlying skin and that the directionality of these processes, like that of the basal cryptic lamellipodia, is dependent on Fgf signaling. Furthermore, we show that mechanically removing the skin prevents PLLp migration, which is subsequently recovered when the skin heals back over

the PLLp. Loss of the overlying skin disrupts polarization of both superficial and basal migratory processes. Embedding skin-removed embryos in Matrigel partially restores both superficial and basal protrusions, but only the polarization of the basal protrusions are recovered, and the primordium remains unable to migrate. Taken together, these data suggest that the PLLp coordinates collective migration by extending lamellipodia both basally, against the underlying tissue, and superficially, against the skin, and that the presence of the overlying skin is essential for PLLp migration.

## Results

We imaged the morphology of individual cells in the PLLp by transplanting cells labeled with a membrane localized EGFP driven by the *cldnb* promoter (Tg(*cldnb:lyn-egfp*)) into *Tg(cldnb:lyn-mscarlet)* transgenic embryos. The resulting chimeric embryos contained mosaically labeled primordia, with a small percentage of donor cells expressing membrane-localized EGFP while the remaining host-derived cells expressed membrane-localized mScarlet (*Figure 1A*).

In most cases, cells in the trailing domain had the characteristic shape of cells incorporated into an epithelial protoneuromast – a basally positioned cell body and a tightly constricted apical domain that connects with the apical domains of other protoneuromast cells to form a microlumen (*Durdu et al., 2014*). In a smaller number of cases, cells in the leading mesenchymal-like domain were labeled. These cells had a flat morphology without any clear apical-basal polarity and extended numerous membranous protrusions against the underlying tissue. However, we also observed cells whose cell bodies lay above the level of the protoneuromast apical constrictions. In some cases, these cells were clearly connected to the apical constriction of a nearby protoneuromast, and they extended across the top of the PLLp and wrapped around the lateral sides (*Figure 1B,C*). In other cases, these superficial cells had no apparent direct connection to the protoneuromast apical constriction, often lying between protoneuromasts. During the course of a timelapse video, we observed these cells extending multiple broad, flat protrusions reminiscent of lamellipodia (*Figure 1D–F*, *Figure 1—video 1*). These protrusions were extended closely apposed to the basal surface of the skin, and their similarity to basal cryptic lamellipodia suggested that they might also contribute to the migration of the PLLp.

To further characterize this superficial population, we performed high-resolution imaging of the PLLp in *Tg(cldnb:lyn-egfp);TgBAC(cxcr4b:h2a-mcherry)* double transgenic fish, in which both the membranes and nuclei of the lateral line primordium are labeled, using both DiSPIM to generate image volumes with isotropic resolution (*Figure 2A–C*), and Airyscan super-resolution confocal microscopy (*Figure 2D*). While the cells that make up the core of the protoneuromast have a basally positioned nucleus, the nuclei of these overlying cells were above the level of the apical constrictions (arrowheads in *Figure 2B,C*), and directly underneath the skin (*Figure 2D*). Slicing this image along the long axis of the PLLp (red dashed line in *Figure 2A*) shows this population of cells occupying a position above the apical constrictions of the protoneuromast cells (*Figure 2B*). Likewise, slicing this image across the cross-section of the PLLp (blue dashed line in A) revealed a series of nuclei occupying a superficial and circumferential position around the apical side of the protoneuromast (*Figure 2C*).

To determine the distribution of these cells along the anterior-posterior extent of the PLLp, we mapped the position of all cells whose nuclei were directly apposed to the overlying skin (red in *Figure 2E,F*), and which resided above the level of the closest protoneuromast apical constriction (blue in *Figure 2E,F*). We then compared their distribution with the distribution of all the cells of the PLLp (gray in *Figure 2E,F*) in both single primordia (for example see *Figure 2E*), or when PLLp were aligned based on the position of the most leading protoneuromast and the positions aggregated (*Figure 2F*). These data show that these cells lie above the fully formed protoneuromasts in the PLLp, but their frequency decreases toward the back of the PLLp, where the trailing-most protoneuromast is preparing to deposit. It should be noted that the apparent lack of these cells in the leading domain is a consequence of our selection criteria, where leading cells, typically flatter and lying closer to the underlying migratory substrate, are excluded. Segmentation and reconstruction of these cells showed that they formed a broad 'sheath' that covered the apical side of the PLLp protoneuromasts, forming a layer between the apical constrictions of the protoneuromast and the overlying skin cells. For illustration, *Figure 2G,H* and *Figure 2—video 1* show a 3D rendering of a

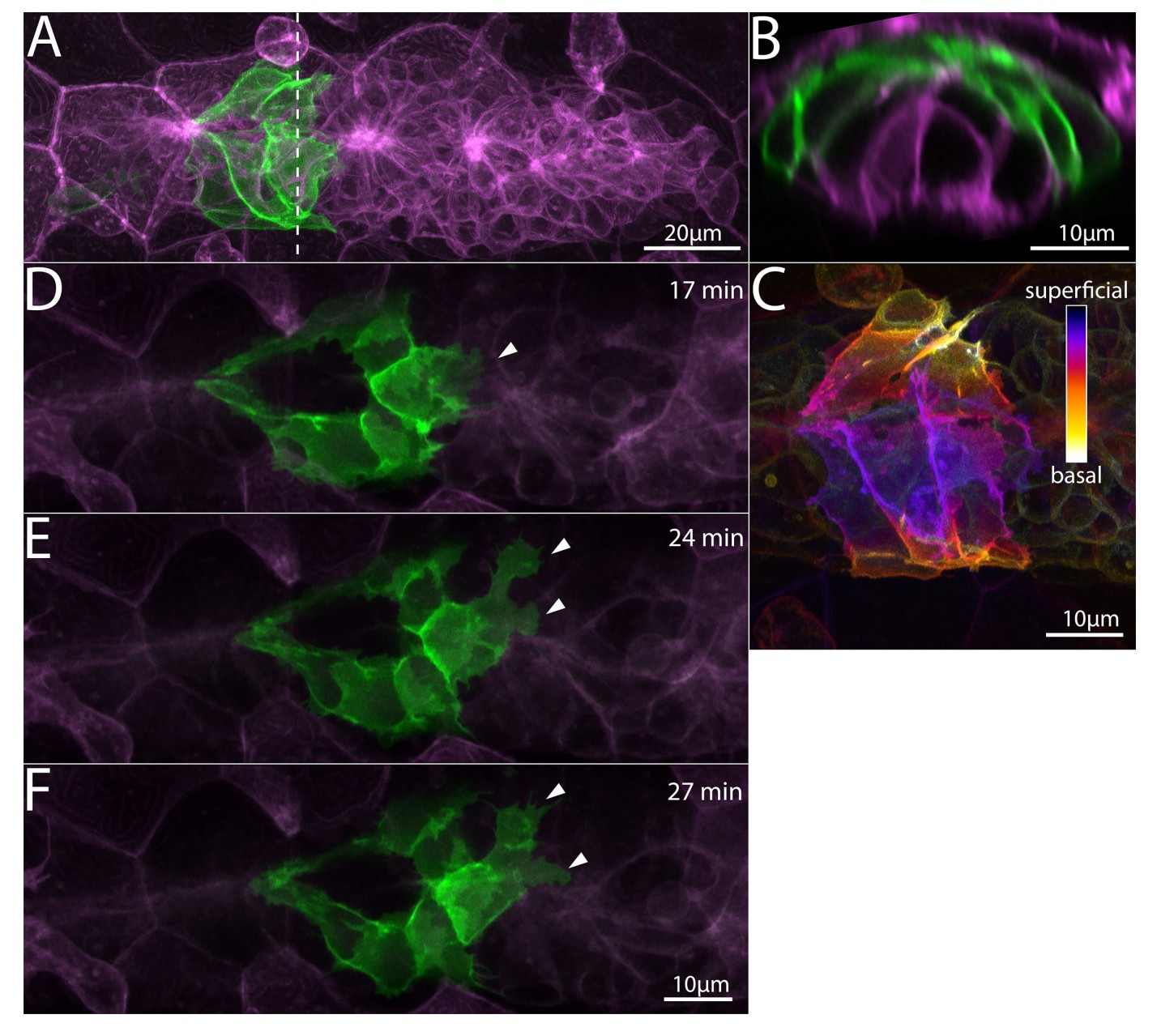

**Figure 1.** Superficial cells in the Zebrafish Posterior Lateral Line primordium. (**A**) Z-projection of cells from a *Tg(cldnb:lyn-egfp)* embryo (green) transplanted into a *Tg(cldnb:lyn-mscarlet)* (magenta) embryo. Dashed line indicates the position of the transverse section in B. (**B**) Transverse section of A showing position of green transplanted cells. (**C**) Superficial-Basal depth coding of the *Tg(cldnb:lyn-gfp)* cells shown in A and B. (**D–F**) Frames from a timelapse video (*Figure 1—video 1*) showing superficial membrane protrusions adjacent to the skin (arrowheads). All panels are processed Airyscan super-resolution images and represent top views of the primordium. Scale bars for D, E same as F.

The online version of this article includes the following video for figure 1:

**Figure 1—video 1.** Related to *Figure 1*.

https://elifesciences.org/articles/58251#fig1video1

fully segmented PLLp with this population colored in magenta, and the remaining cells are colored in green.

We call these cells which lie in-between the skin and the protoneuromasts superficial primordium cells to reflect their superficial position in the primordium, and to distinguish them from the leading

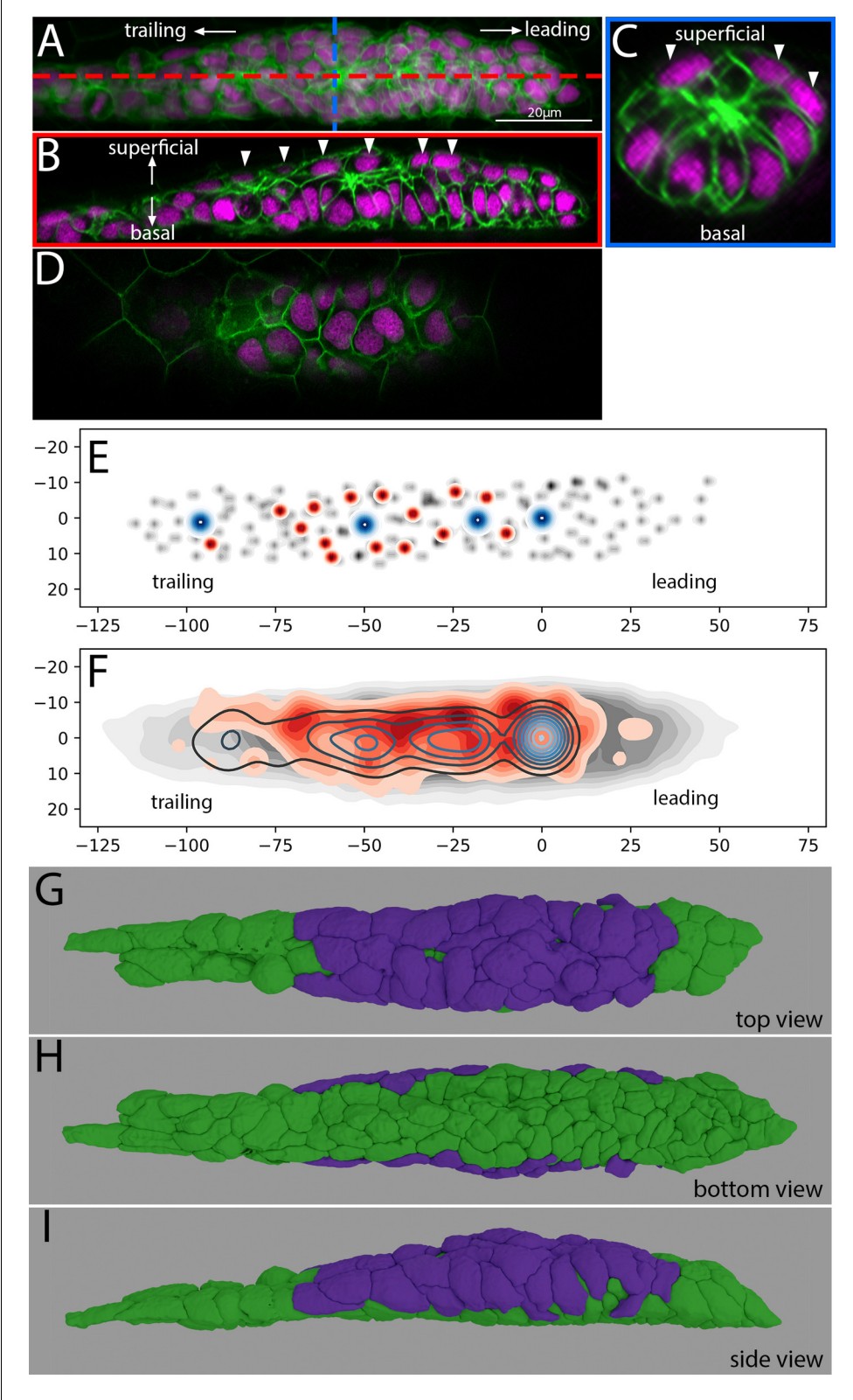

**Figure 2.** Location and morphology of superficial cells in the Zebrafish Posterior Lateral Line Primordium. (**A**) Z-projection of the PLLp at ~32 hpf from a single deconvolved DiSPIM volume. Red and blue dashed lines indicate the position of cross-sections in B and C, respectively. Membranes are indicated in green and nuclei in magenta. (**B**) Length-wise section of the PLLp at the position indicated in by the red line in A. Arrowheads indicate superficial nuclei. (**C**) Cross-section of the PLLp at the position indicated by the blue line in A. Arrowheads indicate superficial nuclei. (**D**) Single

*Figure 2 continued on next page*

*Figure 2 continued*

processed Airyscan super-resolution confocal slice adjacent to the skin showing superficial nuclei. (**E**) Schematic of nuclear position in a single PLLp taken from an Airyscan-processed confocal stack. Gray dots indicate all PLLp nuclei, red dots indicate the position of superficial nuclei (see text for details), and blue dots indicate the position of protoneuromast apical constrictions. (**F**) Average of the positions of nuclei in n = 10 PLLp taken from Airyscan confocal stacks. PLLp were aligned based on the position of the leading-most protoneuromast. Gray indicates the position of all nuclei, red filled topo-lines indicate the position of superficial nuclei and blue open topo-lines indicate the position of protoneuromast apical constrictions. X and Y axes in E, F are distance in microns from the center of the first formed protoneuromast, located at 0,0. E,F are top view of the PLLp as in panel G. (**G, H, I**) 3D reconstruction of a DiSPIM volume of a ~32 hpf PLLp showing the superficial cells in magenta and the remaining cells in green in top (**G**), bottom (**H**), and side (**I**) views.

The online version of this article includes the following video and figure supplement(s) for figure 2:

**Figure supplement 1.** 3D rendering of example cells from cell segmentation shown in *Figure 2*.

**Figure supplement 2.** Surface area contacts and position of superficial cells in the Zebrafish Posterior Lateral Line primordium.

**Figure 2—video 1.** Related to *Figure 2*.

https://elifesciences.org/articles/58251#fig2video1

**Figure 2—video 2.** Related to *Figure 2—figure supplement 1*.

https://elifesciences.org/articles/58251#fig2video2

cells and the protoneuromast cells. These cells, like all cells in the PLLp, are labeled with the *claudinb* promoter, typically thought to reflect their epithelial lineage. However, these cells are clearly morphologically distinguishable from the cells that make up a protoneuromast as they are not obviously apico-basally polarized, and lack the clear apical constrictions of protoneuromast cells. Examples of leading, protoneuromast, and superficial cells can be seen in *Figure 2—figure supplement 1*.

From our segmentation, we can approximate the extent of the external surface contact area of the PLLp by cell type. We manually divided the PLLp into three cell types: leading cells, which occupy the very leading domain of the PLLp (13 cells), protoneuromast cells, cells which appeared apico-basally polarized and terminated in an apical constriction at the center of a protoneuromast (63 cells) and superficial cells, which contacted the overlying skin and were positioned superficial and lateral to the protoneuromast cells (45 cells). In this analysis, we ignored trailing cells which were depositing/deposited as interneuromast cells, as they are unlikely to contribute meaningfully to collective migration of the primordium.

In this segmented data set, we can quantify the external surface area of cells by measuring the number of surface pixels not in contact or close proximity to pixels belonging to other PLLp cells. In this reconstruction, of a total external surface area contact size of ~4400 square microns, approximately half (~2350 $\mu m^2$) is made up of superficial primordium cells (exclusively making contact laterally and superficially) while protoneuromast cells (exclusively contacting the basal surface) make up only 1300 $\mu m^2$. Leading cells making up the remaining 750 $\mu m^2$ (*Figure 2—figure supplement 2A*). While almost all cells have some external contact, the superficial contacts are dominated by superficial cells, while the basal contacts are dominated by protoneuromast cells (*Figure 2G–I*, *Figure 2—figure supplement 2A*). In addition, while there are fewer superficial cells (43 vs 64), the average external contact area of superficial cells (54 $\mu m^2$) is generally larger than the external surface area for protoneuromast cells (21 $\mu m^2$), despite protoneuromast cells having more total surface area (*Figure 2—figure supplement 2A,B*). This is primarily due to the flatter morphology of superficial cells when compared to the elongated, apically constricted nature of protoneuromast cells (see *Figure 2—figure supplement 1*), where the basal 'feet' occupy a relatively smaller proportion of the cell surface area. This greater packing of the basal surfaces of the protoneuromast cells when compared to the extended, flat morphology of the superficial cells can be appreciated in renderings in *Figure 2G,H*, *Figure 2—video 1*, *Figure 2—figure supplement 2A*. These approximate proportions were similar both within a single embryo over time, and between multiple embryos analyzed (data not shown).

To assess the location of this population over time, we tracked the nuclei of all cells in a migrating primordium over the course of two hours. In this video, while there was significant movement of superficial nuclei around the periphery of the primordium (see *Figure 2—video 2*), cells in the superficial population maintained a more superficial and peripheral position, compared to the more centrally and basally located nuclei of the cells making up the protoneuromast, (*Figure 2—figure*

supplement 2C–I). We did not observe any instance of superficial cells internalizing and integrating into a protoneuromast.

The presence of a cell population with minimal or nonexistent basal contact with the underlying tissue but significant surface area contact with the overlying skin suggested that these cells might

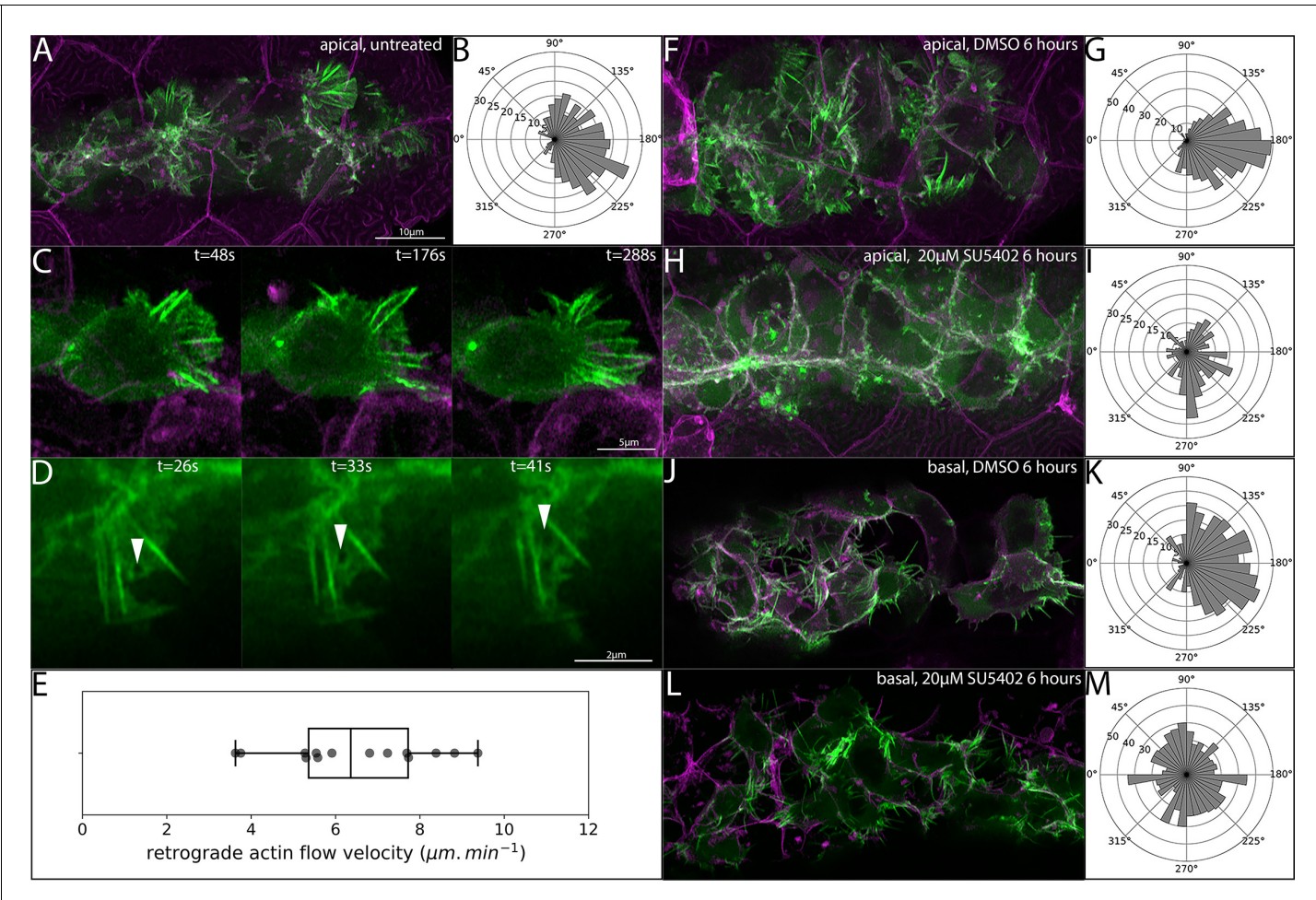

**Figure 3.** Orientation of superficial-cell Actin protrusions in untreated and Fgf-inhibited embryos. (A) Maximum intensity projection of a confocal stack showing LifeAct-Citrine-positive projections (green) within 5 µm of the skin (magenta). (B) Quantification of the direction of superficial actin protrusions from timelapse videos. 180˚ (right) indicates the normal direction of PLLp migration. (C) Frames from a timelapse video showing a single *TgBAC(cxcr4b: lifeact-citrine)* cell adjacent to the overlying skin with polarized protrusive activity. (D) Series of frames from a timelapse video showing retrograde flow of LifeAct-Citrine in a superficial protrusion. (E) Boxplot of quantification of retrograde flow velocity in superficial protrusions. (F) Superficial LifeAct-Citrine-positive protrusions after 6 hr of treatment with DMSO. (G) Directionality of superficial protrusions after DMSO treatment. (H) Superficial LifeAct-Citrine-positive protrusions after 6 hr of treatment with 20 µM SU5402. (I) Directionality of superficial protrusions after SU5402 treatment. (J) Basal protrusions in *TgBAC(cxcr4b:lifeact-citrine)* transplants after 6 hr of treatment with DMSO. (K) Directionality of basal protrusions after DMSO treatment. (L) Basal protrusions in *TgBAC(cxcr4b:lifeact-citrine)* transplants after 6 hr of treatment with 20 µM SU5402. (M) Directionality of basal protrusions after SU5402 treatment. Scale bar for F,H,J,L is the same as A. All image panels are processed Airyscan super-resolution confocal images.

The online version of this article includes the following video(s) for figure 3:

**Figure 3—video 1.** Related to *Figure 3*.
https://elifesciences.org/articles/58251#fig3video1
**Figure 3—video 2.** Related to *Figure 3*.
https://elifesciences.org/articles/58251#fig3video2
**Figure 3—video 3.** Related to *Figure 3*.
https://elifesciences.org/articles/58251#fig3video3
**Figure 3—video 4.** Related to *Figure 3*.
https://elifesciences.org/articles/58251#fig3video4

contribute to the migration of the PLLp. Imaging suggested the presence of broad membrane protrusions from these cells which were oriented in the direction of migration (*Figure 1D–F*, *Figure 1—video 1*). To further examine these protrusions, we performed live imaging using Airyscan super-resolution microscopy in embryos where a BAC containing the *cxcr4b* regulatory elements was used to drive *lifeact-citrine*, which labels F-Actin (*Colak-Champollion et al., 2019*; *Riedl et al., 2008*), in the PLLp. The distribution of LifeAct-Citrine revealed the presence of transient Actin fibers extending within broader membrane protrusions, reminiscent of lamellipodia (*Figure 3A*, *Figure 3—video 1*). These protrusions were even more apparent when we imaged isolated cells by transplanting cells from *TgBAC(cxcr4b:lifeact-citrine);Tg(cldnb:lyn-mscarlet)* double transgenic embryos into *Tg(cldnb:lyn-mscarlet)* embryos (*Figure 3C*, *Figure 3—video 2*). Quantification of the direction of these protrusions in Airyscan super-resolution timelapse videos of the superficial surface of the PLLp showed that they were highly polarized, extending in the direction of migration (*Figure 3B*), with an average length of ~2.4 μm (green in Figure 6G), similar in orientation and length to previously described polarized migratory protrusions that extend from the basal aspect of cells that make up a protoneuromast (average length ~1.9 μm, green in Figure 6N). In contrast, the apical surface of most cells in a protoneuromast is tightly constricted, and we did not observe significant apical protrusive activity from these cells.

In our timelapse videos of *Tg(cxcr4b:lifeact-citrine)* transgenic embryos, we noticed signs of significant retrograde actin flow in these protrusions. In migrating cultured cells, the rate of retrograde actin flow in lamellipodial protrusions has been shown to correlate with the traction force exerted by the migrating cell (*Gardel et al., 2008*). To assess the rate of retrograde actin flow, we took high-speed super-resolution Airyscan videos at 1-s intervals and used the movement of inhomogeneities in the signal to assess the rate of retrograde actin flow in these protrusions (*Figure 3—video 3*, example shown in *Figure 6—figure supplement 1*). *Figure 3D* shows one such inhomogeneity flowing backwards from the lamellipodial tip toward the cell body (arrowheads in *Figure 3D*). Quantification of the flow showed an average retrograde actin flow speed of 6.5 $\mu m.min^{-1}$ with a standard deviation of 1.8 $\mu m.min^{-1}$ (*Figure 3E*).

The orientation of migratory protrusions on the basal surface of the PLLp is known to be dependent on Fgf signaling, and directional polarization is lost when Fgf signaling is inhibited (*Lecaudey et al., 2008*). To test whether the orientation of the superficial protrusions, like those on the basal side of the PLLp, is also dependent on Fgf signaling, we treated embryos with either 20 μM of SU5402 or DMSO for 6 hr and measured the orientation of the superficial protrusions. After 6 hr of treatment with DMSO, superficial protrusions remained robustly polarized in the direction of migration (*Figure 3F,G*, *Figure 3—video 4*, upper left panel). However, this polarization was lost after Fgf inhibition by SU5402 treatment (*Figure 3H,I*, *Figure 3—video 4*, upper right panel), suggesting that, like the basal protrusions, the directionality of these apical protrusions was also dependent on Fgf signaling. For comparison, we performed the same analysis on basal protrusions. However, since the densely packed cell membranes comprising the basal surface of the PLLp make quantification of these basal protrusions challenging, we performed these experiments in chimeric embryos where we transplanted *TgBAC(cxcr4b:lifeact-citrine);Tg(cldnb:lyn-mscarlet)* donor cells into *Tg(cldnb:lyn-mscarlet)* host embryos. As expected, in DMSO-treated embryos the basal protrusions were primarily oriented in the direction of migration (*Figure 3J,K*, *Figure 3—video 4*, lower left panel). However, after 6 hr of treatment with SU5402, this polarity was completely abolished (*Figure 3L,M*, *Figure 3—video 4*, lower right panel).

These experiments demonstrate the existence of a population of cells that make significant contact with the overlying skin during migration. Furthermore, our analysis show that these cells extend protrusions reminiscent of lamellipodia against the skin, and that the orientation of these protrusions, like that of the basal cryptic lamellipodia, is sensitive to Fgf inhibition. This suggested a potential role for the overlying skin as a substrate for migration of the PLLp. To test whether the skin is necessary for PLLp migration, we removed the skin overlying the PLLp using a tungsten needle and then imaged the resulting behavior of the PLLp using regular (diffraction-limited) confocal microscopy. We performed these experiments both in *tg(claudinb:lyn-egfp)* embryos, in which the periderm (outer skin layer) is labeled with membrane localized Gfp, and in *tg(claudinb:lyn-gfp; krt4:dsred)* double transgenic embryos, in which both the periderm and basal layers of the skin are additionally labeled with cytoplasmically localized DsRed (*O'Brien et al., 2012*). Immediately following skin removal, the PLLp ceased migration and remained stationary.

In most cases, after skin removal, the skin rapidly healed over the PLLp with little to no damage to the PLLp itself (based on the minimal appearance dying cells). In these cases, after skin regrowth, the PLLp recovered normal migration and continued migrating along the length of the embryo (*Figure 4—video 1*). During this subsequent migration neuromast deposition appeared normal, and the primordium reached the tip of the tail after a slight delay (*Figure 4—figure supplement 1*). In a minority of cases, where large patches of skin covering a significant fraction of the trunk were removed such that the skin could not heal over the PLLp in the time course of our videos, we observed the PLLp for a period of several hours. In these cases, the PLLp did not recover forward migration, and the PLLp cells eventually died.

*Figure 4A—C* shows three still frames from a representative time-lapse video taken while the skin healed over the PLLp. Initially the PLLp is stationary, having stopped forward migration. The skin (magenta) is healing toward the PLLp (*Figure 4A*). Twenty minutes later, the skin has made contact with the PLLp, and the morphology of the leading cells has changed, becoming more stretched out, reminiscent of their morphology in intact primordia (*Figure 4B*). By 97 min, the skin has healed over the PLLp and normal forward migration has robustly resumed (*Figure 4C*). This dramatic behavior can be seen in kymographs of both the green PLLp membranes and the magenta *krt4*:DsRed-positive epidermal cells (*Figure 4D–F*). Initially, the PLLp is stationary, while uncovered by the skin (compare *Figure 4E,F*). However, as the skin heals over the PLLp, robust and continuous forward migration is recovered, as shown by diagonal lines in the kymograph (*Figure 4D–F*). We removed the skin from >10 independent embryos and, in all cases, migration was abolished until the skin healed over the PLLp.

To quantify this migratory behavior, we crossed *Tg(claudinb:lyn-egfp; krt4:dsred)* fish to *TgBAC (cxcr4b:h2a-mcherry)* fish to visualize the deeper basal cells and more superficial periderm of the skin, as well as both the membranes and nuclei of the PLLp. We then removed the skin overlying the PLLp and tracked the movement of randomly selected cells distributed throughout the PLLp for each of three replicate embryos. For each cell, the time at which the skin covers the position of that individual cell was marked, and the average velocity in the normal direction of migration (along the rostral-caudal axis of the embryo) before and after this point was calculated. *Figure 4G* shows the paired measurements for each cell, with the top row representing the average velocity of cells before skin contact and the bottom row representing the average velocity after skin contact. Consistent with the results from bulk analysis of movement using kymographs, there was a dramatic increase in forward migration after skin contact. The velocity of cells before skin contact was distributed around 0, suggesting non-directional movement, and the velocities after skin contact are clustered around a mean of 0.82 $\mu$m.min$^{-1}$, close to the normal migration speed of a primordium. Aggregating the frame-to-frame velocities shows that the velocity before skin contact is normally distributed around a value of approximately zero (mean = 0.067 $\mu$m.min$^{-1}$, standard deviation = 0.51 $\mu$m.min$^{-1}$), whereas the velocities after skin contact are normally distributed around approximately 0.8 $\mu$m.min$^{-1}$ (mean = 0.807 $\mu$m.min$^{-1}$, standard deviation = 0.648 $\mu$m.min$^{-1}$). Taken together, these data suggest a profound inability of the PLLp to migrate without overlying skin, which is completely reversed after skin regrowth over the PLLp.

Interestingly, removal of the skin was associated with a significant increase in the average height of the primordium. Paired measurements of the average height of the PLLp before and after skin removal showed a ~ 18% increase in the height of the PLLp associated with removal of the overlying skin (*Figure 4H*), suggesting that the skin does in fact provide some compression to the PLLp.

We hypothesized that removing the skin caused a failure of the overlying cells to extend robust migratory protrusions and that this phenomenom contributes to the failure of collective migration. To assess this, we crossed *TgBAC(cxcr4b:lifeact-citrine)* fish to *Tg(cldnb:lyn-m-scarlet)* fish to generate double-transgenic embryos in which both Actin and membranes were labeled. The skin over the PLLp was then removed and the resulting Actin dynamics imaged using Airyscan super-resolution confocal microscopy.

After this manipulation, we noticed a dramatic change in the behavior of the overlying cells. In PLLp which were not covered by skin, we saw almost no polarized Actin-rich protrusions of the kind observed extending from overlying cells in intact embryos. Instead, cells appeared disorganized and rapidly extended and retracted protrusions reminiscent of membrane blebs. High-resolution imaging of these bleb-like structures showed that they initiate as a rapid membrane expansion devoid of cortical actin (*Figure 5A–E*, *Figure 5—video 1*). As the bleb expands, cortical Actin is recruited to the

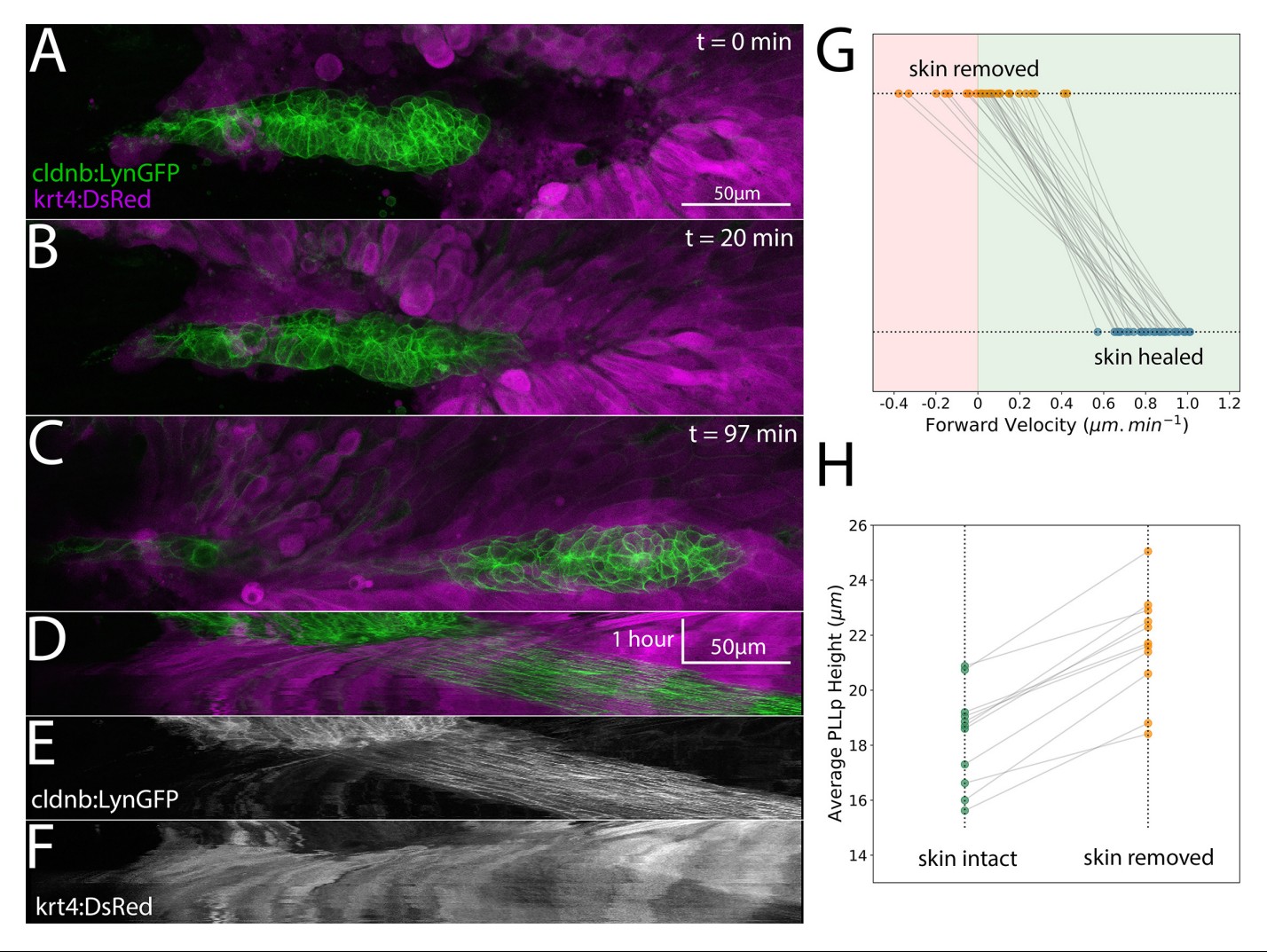

**Figure 4.** Posterior Lateral Line primordium migration after removal of the overlying skin. (**A–C**) Frames from a timelapse video showing the skin (basal and periderm layers labeled by *krt4:dsred*) in magenta and the PLLp in green after skin removal and during subsequent healing. (**D**) Overlay of kymograph showing movement of the PLLp (green) and healing of the skin over the PLLp (magenta) along the migration course of the PLLp shown in A-C. (**E**) Kymograph of the PLLp alone. (**F**) Kymograph of the skin alone. (**G**) Quantification of the average migration speed of individual cells (n = 28) in three independent primordia with skin removed (orange dots) and after skin healing (blue dots). Red background indicates rostral movement, green background indicates caudal movement, p=2e$^{-17}$ (paired sample t-test). (**H**) Quantification of average PLLp height before (green dots) and after (orange dots) skin removal. p=6e$^{-7}$ (paired sample t-test, n = 11). Each gray line in G and H connects two paired measurements of the same PLLp or cell before and after manipulation.

The online version of this article includes the following video and figure supplement(s) for figure 4:

**Figure supplement 1.** Image of a *Tg(cldnb:lyn-egfp)* embryo where skin had been removed over the PLLp at ~32 hpf (approximately equal in size and position as indicated by the white dashed box) and migration had been allowed to proceed until 54hpf.

**Figure 4—video 1.** Related to *Figure 4*.

https://elifesciences.org/articles/58251#fig4video1

newly expanded membrane and can be detected by an increase in the LifeAct-Citrine signal, while the Lyn-mScarlet membrane marker remains constant in intensity (*Figure 5E*). After a short period, these blebs are retracted into the cell.

Because the regrowth of the skin over the PLLp is associated with a rapid and robust recovery of forward migration, we imaged Actin dynamics in the PLLp during skin regrowth in *TgBAC(cxcr4b: lifeact-citrine)*; Tg(*cldnb:lyn-m-scarlet*) embryos using Airyscan super-resolution confocal microscopy (*Figure 5F–I*, *Figure 5—video 2*). After skin removal – but before skin regrowth over the PLLp – we

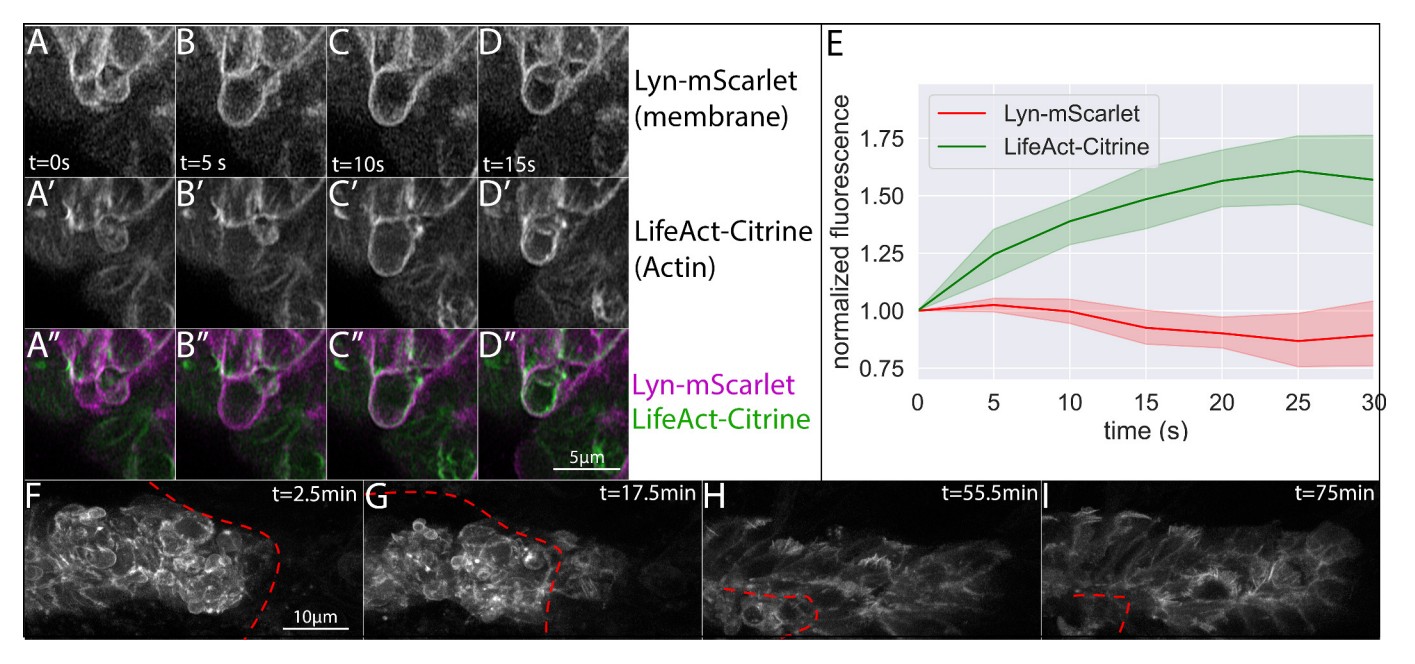

**Figure 5.** Morphology of superficial cells after removal and during regrowth of the overlying skin. (**A–D**) Frames from a timelapse video showing superficial bleb formation and retraction after skin removal. (**A-D**) show lyn-mScarlet, marking membranes, A'-D' show LifeAct-Citrine, marking F-actin and A''-D'' show the merge of both channels. Each panel (**A-D**) represents an interval of 5 s (**E**) Quantification of membrane (lyn-mScarlet) and Actin (LifeAct-Citrine) fluorescence intensity along bleb edge over time for n = 8 blebs. (**F-I**) Frames from a time-lapse video showing superficial LifeAct-Citrine during skin healing. Skin edge position is shown by the dashed red line, and time is indicated in the upper right corner. All images are processed Airyscan superresolution confocal images.

The online version of this article includes the following video(s) for figure 5:

**Figure 5—video 1.** Related to *Figure 5*.
https://elifesciences.org/articles/58251#fig5video1
**Figure 5—video 2.** Related to *Figure 5*.
https://elifesciences.org/articles/58251#fig5video2

observed rapid membrane blebbing of superficial cells, as described above. However, as the skin heals over the PLLp, cells undergo a dramatic morphological change. The bleb-type morphology of overlying cells is abolished and cells begin to extend actin-rich protrusions in the direction of migration, reminiscent of those observed in superficial cells in intact, unperturbed embryos. This transition was so rapid that over the course of our timelapse video, we could simultaneously observe blebs from regions of the PLLp not yet covered by skin alongside actin-rich rod-like protrusions from cells that had been covered by the regrowing skin. Eventually, the entire PLLp is again covered by the skin, superficial cell protrusive activity is restored, and normal forward migration of the PLLp resumes.

These data suggested a rapid recovery of superficial cell protrusive activity after skin regrowth. To quantify this change, we performed high-resolution imaging of Actin dynamics in *TgBAC(cxcr4b: lifeact-citrine);Tg(cldnb4.2:lyn-mscarlet)* embryos under three conditions: unperturbed embryos in which the skin has not been removed, embryos in which the skin was removed over the PLLp, and embryos in which the skin had been removed and had subsequently healed over the PLLp. *Figure 6A–F* shows the results of this experiment, and examples of protrusions can be seen in *Figure 6—figure supplement 1A*. As expected, embryos where the skin had not been removed showed robust protrusions oriented in the direction of migration (*Figure 6A,B*, *Figure 6—video 1*, upper left panel), with an average length of ~2.4 um (*Figure 6G*, green). When the skin was removed, we again observed a profound loss of superficial Actin-rich protrusions, which were replaced by superficial membrane blebs of the type shown in *Figure 5A–D* (*Figure 6C*, *Figure 6—video 1*, upper middle panel). The few protrusions that remained were both shorter (*Figure 6G*,

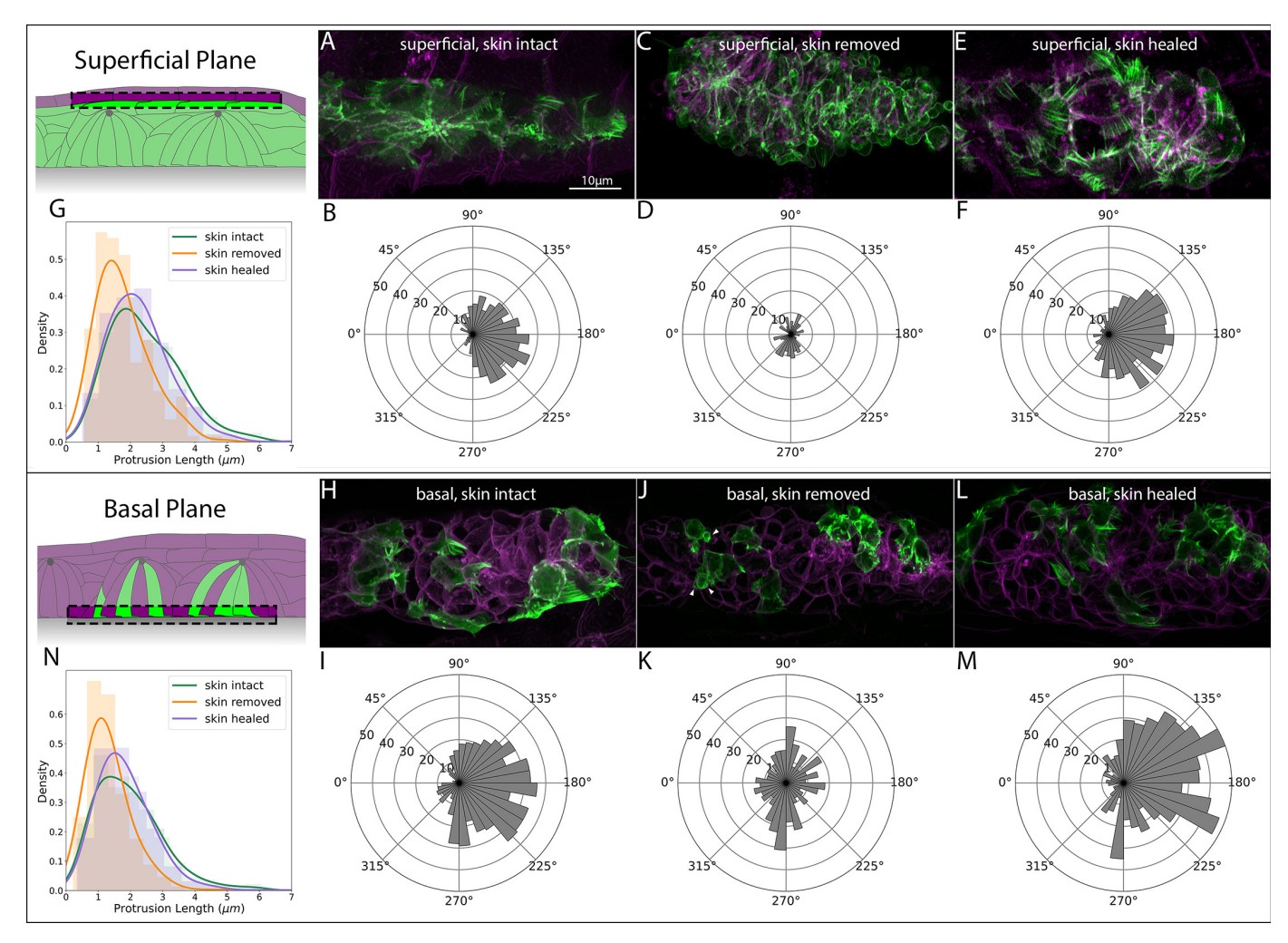

**Figure 6.** Superficial and basal protrusion morphology and directionality before, during and after skin removal and regrowth. (A) Superficial LifeAct-Citrine-positive protrusions in embryos with intact skin. (B) Quantification of superficial protrusion directionality in embryos with intact skin. Panel K reflects same data as *Figure 3B*. (C) Superficial LifeAct-Citrine after removal of the overlying skin. (D) Quantification of superficial protrusion directionality in PLLp with overlying skin removed. (E) Superficial LifeAct-Citrine after removal and regrowth of the overlying skin. (F) Quantification of superficial protrusion directionality in PLLp with overlying skin removed and allowed to regrow. (G) Histograms and kernel density estimate (kde) plots of superficial-cell protrusion length for embryos with skin intact (green), skin removed (orange) and skin removed and allowed to heal (purple). (H) Transplanted cells showing basal LifeAct-Citrine-positive protrusions in embryos with intact skin. (I) Quantification of basal protrusion directionality in embryos with intact skin. (J) Transplanted cells showing basal LifeAct-Citrine after removal of the overlying skin. (K) Quantification of basal protrusion directionality in PLLp with overlying skin removed. (L) Basal LifeAct-Citrine after removal and regrowth of the overlying skin. (M) Quantification of basal protrusion directionality in PLLp with overlying skin removed and allowed to heal. (N) Histograms and kde plots of basal protrusion length for embryos with skin intact (green), skin removed (orange) and skin removed and allowed to heal (purple). All images are maximum-intensity projections of confocal stacks within 5 µm of the skin (for superficial protrusions) or the basal surface of the PLLp (for basal slices). All image panels are processed Airyscan super-resolution confocal images. Scale bar for (A), (C), (E), (H), and (J) are same as A. Schematics in each panel show the approximate area of the PLLp being imaged (dashed boxes).

The online version of this article includes the following video and figure supplement(s) for figure 6:

**Figure supplement 1.** Examples of protrusions measured for direction and length quantification in *Figures 3*, *6* and *7*.

**Figure 6—video 1.** Related to *Figure 6*.

https://elifesciences.org/articles/58251#fig6video1

compare orange to green, mean skin intact = 2.45 µm, mean skin removed = 1.76 µm) and no longer oriented in the direction of migration (*Figure 6D*). However, as suggested by the timelapse analysis above, after the skin healed over the PLLp, superficial protrusions were again observed (*Figure 6E*, *Figure 6—video 1*, upper right panel), their orientation was again strongly polarized in the direction of migration (*Figure 6F*) and their length had recovered to almost unperturbed levels (*Figure 6G*, compare green and purple, mean intact = 2.45 µm, mean post-healing = 2.26 µm).

Although the absence of these directional superficial cell protrusions was expected after removal of the overlying skin, we were surprised by the profound loss of migratory ability resulting from this manipulation, given that the basal surface of deeper protoneuromast cells were presumably still in contact with the underlying tissue. We wondered whether removal of the skin resulted in a broader loss of migratory protrusions in the cells of the PLLp. To assess this, we repeated the above experiment, this time using embryos in which we transplanted donor cells from *Tg(BACcxcr4b:lifeact-citrine; cldnb:lyn-mscarlet)* into *Tg(cldnb:lyn-mscarlet)* embryos to generate isolated clones in which we could quantify the directionality and length of basal protrusions.

In intact embryos, these basal protrusions, like the superficial cell protrusions, were highly polarized in the direction of migration, although they were on average slightly shorter than superficial cell protrusions (*Figure 6H,I,N*, average length = 1.95 µm, *Figure 6—video 1*, bottom left panel). When the skin was removed, surprisingly, we observed a profound loss of the normal directional orientation of these basal protrusions (*Figure 6J,K*, *Figure 6—video 1*, bottom middle panel), even though many of these cells had no contact with the overlying skin. This failure of directional orientation was accompanied, as with the superficial protrusions, by a decrease in the average length of the protrusions, (*Figure 6N*, compare green to orange, mean intact 1.95 µm, mean skin removed = 1.32 µm) and by the appearance of membrane blebs on the basal surface of these cells (arrowheads in *Figure 6J*). However, neither the loss of rod-like Actin-rich protrusions nor the appearance of membrane blebs were as dramatic on the basal surface as on the superficial surface.

When we imaged these protrusions after skin regrowth (*Figure 6L*, *Figure 6—video 1*, bottom right panel), we observed the same dramatic recovery of basal protrusions. The polarization of the basal protrusions in the direction of migration was recovered (*Figure 6M*) and their length was similar to that of protrusions in unperturbed embryos (*Figure 6N*, compare purple to green, mean intact = 1.95 µm, mean post healing = 1.83 µm).

Given the failure to observe superficial cell protrusions in primordia where the skin had been removed and embryos re-embedded in agarose, which is expected to provide some mechanical confinement, we wondered whether interactions with the extracellular matrix associated with overlying skin cells were necessary for these protrusions. To test this, we removed the skin over the PLLp in 36hpf embryos and embedded these embryos for imaging in 20 mg/mL Matrigel, a gelatinous artificial 3D medium derived from extracellular matrix, rather than agarose. We reasoned that by using undiluted 20 mg/mL Matrigel, we would provide the stiffest possible substrate for primordium migration.

Interestingly, we noted that healing of the skin was significantly inhibited in Matrigel, allowing us to image primordia in skin-free conditions for longer than is possible in agarose. In most cases, primordia either did not migrate, or migrated forward only short distances, less than the length of the primordium, and then stalled again and did not recover migration. Quantification showed that when skinned embryos were embedded in Matrigel, migration speed was significantly slower (~0.2 µm/min) than embryos with skin intact (~0.8 µm/min), suggesting that Matrigel is unable to completely replace the skin for normal primordium migration. When compared to skinned embryos in agarose, skinned embryos in Matrigel showed a slight, but not statistically significant, increase in migration speed (*Figure 7E*).

Despite this failure to recover migration when skinned embryos are embedded in matrgiel, we observed a profound recovery in the morphology of the superficial cells, which no longer exhibited blebbing behavior, but instead extended actin-rich lamellipodial-like processes, qualitatively similar to those observed in unperturbed embryos (*Figure 7A*). However, when we quantified the directionality of these protrusions, we saw that they were oriented in all directions, unlike the highly polarized protrusions observed in intact embryos (*Figure 7B*). In contrast, when we embedded embryos in which we had removed the skin overlying the PLLp in Matrigel and examined the polarity of basal protrusions, we saw a robust recovery of the polarization of these protrusions (*Figure 7C,D*). This

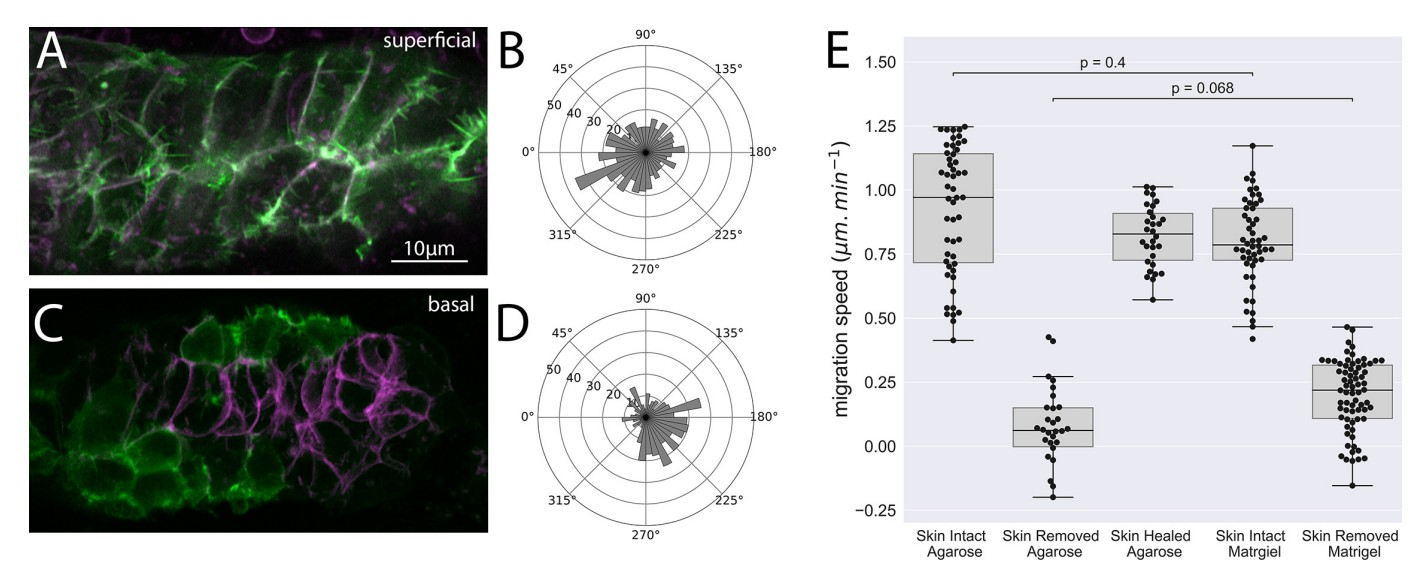

**Figure 7.** Superficial and basal protrusion morphology and directionality after removal of the overlying skin and embedding in Matrigel. (**A**) Superficial LifeAct-positive protrusions (green) and membrane (magenta) in skin-removed embryos embedded in 20 mg/mL Matrigel. (**B**) Quantification of superficial protrusion directionality in embryos embedded in Matrigel (**C**) Basal LifeAct-positive protrusions (green) and membrane (magenta) in skin-removed embryos embedded in 20 mg/mL Matrigel (**D**) Quantification of basal protrusion directionality in embryos embedded in Matrigel. (**E**) Box plots of average cell migration speed (in the direction of normal migration) for embryos with skin intact embedded in 1% agarose, skin removed embedded in 1% agarose, skin removed and regrown embedded in 1% agarose, skin intact in 20 mg/mL Matrigel, and skin removed embedded in 20 mg/mL Matrigel. Dots indicate individual cell speed in the direction of normal migration in at least three embryos. p Values are represent results from ANOVA with multilevel model. Panels (**A, C**) are processed Airyscan super-resolution images.

recovery was striking when compared to the loss of this polarization when skinned embryos were embedded in agarose.

## Discussion

In this study, we examined the Posterior Lateral Line primordium of zebrafish, a well-studied model of collective cell migration that migrates in a channel between the underlying somites and overlying epidermis. We show that, in addition to the previously described basal lamellipodia extended by epithelialized cells in this cluster (**Haas and Gilmour, 2006**), there is an additional population of cells lying superficially, covering these epithelial cells, which also extend migratory processes against the overlying skin. As with the directionality of basal cryptic lamellipodia extended from the epithelial cells that comprise a protoneuromast, the directionality of these superficial processes is abolished when Fgf signaling is inhibited by treatment with SU5402, suggesting that the requirement for Fgf signaling to maintain migration applies to both superficial and basal processes.

Supporting the idea that these migratory processes are in fact lamellipodia, we show retrograde Actin flow in these protrusions toward the cell body at approximately 6.5 µm.min$^{-1}$. Values reported from cultured cells in vitro vary significantly, from 0.12 to 1.5 µm.min$^{-1}$ in the distal lamellipodium of Ptk1 cells (**Gardel et al., 2008**), to 0.8 – 4.5 µm.min$^{-1}$ for spreading mouse embryonic fibroblasts (**Dubin-Thaler et al., 2004**) and 2.5–6 µm.min$^{-1}$ in migrating fish keratinocytes (**Jurado et al., 2005**). Retrograde actin flow in migrating cells in live embryos has not been widely quantified; however, a recent study examining *Drosophila* macrophages (**Yolland et al., 2019**) in vivo showed a retrograde flow rate of approximately 4 µm.min$^{-1}$. Explant cultures of Xenopus spinal cord neurons showed higher rates for neuron growth cones, from 9 to 12 µm.min$^{-1}$ depending on substrate (**Nichol et al., 2016**). Interestingly, when these rates were assessed in vivo, retrograde flow was slower, at around 7 µm.min$^{-1}$, which the authors speculate may be due to stronger F-actin clutching in the in vivo 3D environment. Our value of 6.5 µm.min$^{-1}$ is on the upper end of the values observed in cell culture, but within the bounds of the values reported in vivo.

In vitro, the rate of actin retrograde flow in Ptk1 cells has a biphasic relationship with traction stress on the substrate; Between 0 and 0.6 µm.min$^{-1}$, retrograde flow rate correlates positively with traction stress, but, between 0.6 and 1.8 µm.min$^{-1}$, it correlates negatively (*Gardel et al., 2008*). In the context of these observations in vitro, the high-retrograde flow rate of actin in superficial cell lamellipodia might suggest low traction stress exerted by these PLLp cells. However, we caution that the rates measured in vitro by Gardel et al are generally less than 1.2 µm.min$^{-1}$, significantly lower than our observed value of 6.5 µm.min$^{-1}$ in PLLp superficial cells. Therefore, how this relatively high-retrograde flow rate measured in vivo correlates with traction stress cannot be inferred with certainty.

Typically, the PLLp has been conceptually separated into a leading mesenchymal-like domain, where cells are relatively flat and have no obvious apico-basal polarity, and a trailing epithelial domain, where cells become elongated and adopt a distinct epithelial morphology with apico-basal polarity. Despite this distinction, it has long been recognized that both domains contribute to collective migration, with epithelial cells extending cryptic lamellipodia in the direction of migration. In this study, we have defined the superficial primordium cells as those that occupy a superficial position above the protoneuromasts, and by definition exclude leading mesenchymal-like cells. However, it is likely that, rather than being an entirely separable population, these superficial cells simply represent an extension of leading mesenchymal-like cells over the entire PLLp and that these cells constitute a continuous migratory population. The Gilmour group has recently performed an elegant cell segmentation and morphological analysis study (*Hartmann et al., 2020*) using support vector classifiers to classify cells in the primordium based on morphology as central cells, leading cells, peripheral cells, as well as so-called inter-organ cells (which we suspect are similar to our superficial cells). While this work showed that PLLp cells populations can in fact be separated into morphologically distinct populations, it also showed significant overlap between these populations, with intermediate states common between, for example, leading and peripheral cells, again suggesting some degree of morphological continuity between distinct cell populations. In this context, in addition to conceptually dividing the PLLp along a longitudinal axis into a leading and trailing domain with distinct cellular morphologies, a second axis is radially arranged with apico-basally polarized epithelial cells incorporated into protoneuromasts at the core and non-protoneuromast, less apico-basally polarized cells at the periphery. This is an example of what Blanchard and co-authors have called 'mesoscale heterogeneity' in migrating systems (*Blanchard et al., 2019*).

Although this study has focused on the protrusions extended by these superficial cells against the overlying skin, we also note that this population also appears to cover the lateral edges of the primordium and the cells themselves seem to be relatively motile, often moving relative to the underlying epithelial cells which they cover. This suggests the possibility that migratory activity is not restricted to the deep or superficial surfaces of the primordium but is in fact circumferential. It is important to note that not only must the epithelial cells of the PLLp migrate effectively in this confined space, but while migrating, deeper cells must also undergo a complex series of coordinated morphogenetic events leading to the formation of apically-constricted protoneuromasts. In addition, previous work has shown that the apical constrictions of protoneuromasts house microlumen structures critical for sequestration of signaling molecules (*Durdu et al., 2014*). Damage to these microlumens causes a failure of Fgf signaling and delayed neuromast deposition. In this context, the presence of a layer of superficial cells between the skin and the protoneuromast cells may not only contribute to collective migration of the primordium, it might also contribute to robust formation of the underlying protoneuromasts during migration and protect their apical microlumens from sheer stress caused by migration against the overlying skin.

Consistent with the existence of migratory contacts on the superficial side of the PLLp closely apposed to the overlying skin, we show that when the skin is removed superficial lamellipodia are almost completely abolished and cells extend and retract rapid membrane blebs. The few Actin-rich rod-like protrusions that remain are no longer oriented in the direction of migration and are significantly shorter than those extended from the superficial surface of the PLLp when the skin is intact. Regrowth of the skin restores both the normal length and directionality of these protrusions with very little delay suggesting, again, a rapid switch in migratory ability and morphology between PLLp cells in the absence and presence of overlying skin. It should be noted that in this context the morphology of superficial cells with and without overlying skin is so dramatically different that even though we can in most cases quantify a small number of rod-like protrusions in primordia without

skin, their relationship to the broad, flat lamellipodial protrusions seen in intact embryos is not clear. It may be that removal of the skin completely disrupts lamellipodia formation, and that a distinct class of actin-rod like protrusions are observed with low frequency in this context.

Intriguingly, we observe very similar changes in the basal cryptic lamellipodia of protoneuromast cells when the skin is removed, despite the fact that these cells, in most cases, do not directly contact the overlying skin. While Actin-rich protrusions are still observed at the basal surface of the PLLp these protrusions, like those observed superficially, become shorter and are no longer oriented in the direction of migration. Concurrent with this loss of directionality, we see the appearance of membrane blebs in the basal surface of these cells, although to a lesser extent than is observed superficially. This observation, along with the rapid recovery of directionality and length of basal protrusions after skin regrowth–despite the fact that these cells do not themselves directly contact the skin–raises the possibility that the skin is not only necessary to provide a substrate for the superficial migratory processes, but that it is provides some indirect support to basal migratory processes as well. Whether this is due to mechanical confinement, restriction of signaling molecules, provision of adhesive interactions, or some other unknown function of the overlying skin remains unclear.

While agarose provides some degree of mechanical confinement, it fails to provide other physiologically relevant factors such as adhesion sites. Matrigel is a complex protein mixture secreted by Engelbreth-Holm-Swarm mouse sarcoma cells and is commonly used as a 3D matrix for growing cultured cells. The Youngs modulus of gelled Matrigel can vary over twofold between batches, with typical values falling around 200–400 Pa (*Soofi et al., 2009*). The Youngs modulus of agarose is significantly higher. Although we have not directly measured the mechanical properties of agarose used in this study, a survey of values from the literature suggest that a value on the order of 10–15 kPa, (approximately 20-fold higher than Matrigel) is a reasonable estimate for the concentrations used (see for example *Kazi et al., 2019*).

When we embedded skin-removed embryos in Matrigel, we observed an almost-complete reversion of blebbing from superficial cells. However, this recovery is not associated with polarization of these superficial protrusions and provides only a modest recovery of cell migration speed. This inability to sustain collective migration is consistent with superficial polarized lamellipodia being required for migration. Furthermore, it suggests that while confinement under Matrigel and the potential to interact with its ECM components may be adequate for the formation of superficial cell lamellipodia, their polarization requires additional factors or environmental constraints that Matrigel is unable to provide. In the case of the basal protrusions, while protrusion directionality is not recovered by embedding in agarose and providing confinement alone, the directionality is recovered when embryos are embedded in the comparatively less stiff Matrigel. This argues against mechanical confinement alone as a requirement for polarized protrusion, instead suggesting that some component provided by Matrigel but not by agarose is required for effective migration. One possibility is that Matrigel is more effective at confining signaling molecules essential for migration than is agarose. While the major protein components of Matrigel appear to be extracellular matrix components such as Laminin, Collagen, and Entactin (*Hughes et al., 2010*), several growth factors have also been detected, and this has led to cautions about interpreting the results of experiments performed in Matrigel (*Vukicevic et al., 1992*). However, precisely why basal and not superficial polarity is recovered in Matrigel remains a mystery, and hints at distinct regulatory mechanisms underlying protrusive activity on the basal and superficial surfaces of the PLLp. In this context, it is important to note that the basal surfaces of cells remain in contact with their in vivo substrate, while the superficial surface is in contact with Matrigel. This may in part explain the recovery of protrusive polarization basally but not superficially. The failure to restore normal migration speed when basal, but not superficial, protrusive directionality is restored in this condition suggests that both superficial and basal migratory protrusions are required for robust migration. Future experiments will attempt to elucidate the determinants of the recovery observed when embryos are embedded in Matrigel, but not agarose, as well as the reason for the failure of Matrigel to rescue migration to wild-type levels.

How cells migrate through a confined environment is poorly understood, especially during collective migration. Unconfined cells in 2D culture can migrate by promoting attachment to a substrate through which force can be transmitted. However, in confined systems such as 3D scaffolds, the mechanical properties of the environment itself can force cells into significant contact with their extracellular migratory substrate. The rapid switch between a lamellipodial form of migration to unproductive blebbing and a failure to migrate is the primary consequence of removal of the skin

from the PLLp. A similarly rapid switch between distinct cell morphologies has also been reported in early zebrafish gastrula cells placed in culture (*Ruprecht et al., 2015*). When unconfined in culture, these cells show a characteristic but unproductive blebbing, and fail to migrate. However, upon mechanical confinement in agarose, the cells adopted a pear-shaped morphology with a large stable leading protrusion and became highly migratory. This migration was associated with rapid retro-grade cortical actin flow in the leading edge of the cell. While in this context the stable leading protrusion does not appear to be lamellipodial as are the protrusions we observe in the superficial cells of the primordium, the rapid switch between these two morphologies - unpolarized and unproductive blebbing to a highly polarized migratory protrusion - upon cell confinement is at least superficially similar. Whether or not these superficial similarities reflect deep mechanistic similarities in the mechanisms of migration in these two contexts will require further study.

Cells in confined environments can also employ so-called 'chimneying' mechanisms, where cells can push outwards against the surrounding environment and generate enough force to facilitate migration. Such a mechanism has been suggested to play a role in migration of leukocytes in an adhesion-independent manner (*Malawista and de Boisfleury Chevance, 1997*). A related but mechanistically distinct mechanism, flow-friction-driven force transmission, is hypothesized to transmit intracellular force from the cytoskeleton to the substrate by means of nonspecific friction between the cell and its environment (*Hawkins et al., 2011*; *Ruprecht et al., 2015*). In this context, retro-grade flows of the actomyosin cortex, like those we observe in superficial protrusions in the PLLp, have been suggested to play a role in the generation of motile force (*Paluch et al., 2016*). However, the ability of Matrigel to recover superficial processes suggests that the presence of extracellular-matrix-like adhesion sites is critical for formation of these migratory processes. Furthermore, a recent report by the Nechiporuk group has shown phospho-paxilin immunoreactivity around the lateral and superficial sides of the PLLp, suggesting that a low-adhesion migration mechanism is unlikely *Olson and Nechiporuk, 2020*. However, the fact that in the vast majority of cases (>99%), we were able to remove the skin from the PLLp while retaining attachment of the PLLp to the underlying substrate suggests an asymmetry of adhesion between the superficial and basal surfaces of the PLLp, and that the basal surface of the PLLp is more tightly adhered to the underlying substrate than the superficial surface is to the overlying skin, despite the presence of superficial adhesions. The significance of this to primordium migration remains unclear and is a subject of further study.

Understanding how cells in the developing embryo move through diverse environments is critical for understanding both normal morphogenesis and a number of pathologies. Cell migration has been studied extensively in cell culture in the two-dimensional regime; however, there is growing evidence that 2D models emphasize specific modes and mechanisms of cell migration that might not apply to cells migrating in a three-dimensional confined environment. While multicellular systems acquire behaviors that correspond broadly to those observed in single cells, it is likely that the details about how they acquire those characteristics differ significantly. The development of in vivo model systems to study cell migration at high spatio-temporal resolution, as well as quantitative methods for analyzing such systems will be necessary to extend the study of 3D cell migration from the dish to the animal.

## Materials and methods

**Key resources table**

| Reagent type (species) or resource | Designation | Source or reference | Identifiers | Additional information |
|---|---|---|---|---|
| Genetic reagent (*Danio rerio*) | *Tg(cldnb:lyn-gfp)* | PMID:16678780 | ZFIN ID: ZDB-ALT-060919–2 | |
| Genetic reagent (*Danio rerio*) | *TgBAC(cxcr4b:lifeact-citrine)* | PMID:26818072 | ZFIN ID: ZDB-ALT-160901–3 | |
| Genetic reagent (*Danio rerio*) | *TgBAC(cxcr4b:h2a-mcherry)* | PMID:31386838 | ZFIN ID: ZDB-ALT-180131–1 | |
| Genetic reagent (*Danio rerio*) | *Tg(cldnb:lyn-mscarlet)* | This paper | | See Materials and methods |
| Gnetic reagent (*Danio rerio*) | *Tg(krt4:dsred)* | PMID:22020759 | ZFIN ID: ZDB-ALT-120127–5 | |

*Continued on next page*

*Continued*

| Reagent type (species) or resource | Designation | Source or reference | Identifiers | Additional information |
|---|---|---|---|---|
| Software, algorithm | Allen Institute for Cell Science Segmenter | doi: https://doi.org/10.1101/491035; | 0.1.16.dev4 | Available: https://www.allencell.org/segmenter.html |
| Chemical compound, drug | SU5402 | Tocris | Tocris:3300 | |

## Fish lines and embryo manipulation

Zebrafish embryos were generated by natural spawning, maintained under standard conditions, and staged according to *Kimmel et al., 1995*. The following lines were used: *Tg(cldnb:lyn-egfp)* RRID: ZFIN_ZDB-ALT-060919-2 (*Haas and Gilmour, 2006*), *TgBAC(cxcr4b:lifeact-citrine)* RRID:ZFIN ID: ZDB-ALT-160901–3 (*Fuentes et al., 2016*), *TgBAC(cxcr4b:h2a-mcherry)* RRID:ZFIN ID: ZDB-ALT-180131–1 (*Colak-Champollion et al., 2019*), *Tg(cldnb:lyn-mscarlet)*, *Tg(krt4:dsred)* RRID:ZFIN ID: ZDB-ALT-120127–5 (*O'Brien et al., 2012*). For generation of *Tg(cldnb:lyn-mscarlet)* transgenic fish, the *lyn-mscarlet* (*Bindels et al., 2017*) DNA sequence was codon optimized for zebrafish expression (*Horstick et al., 2015*) and commercially synthesized. This fragment was cloned downstream of the 4.2 kb *claudinb* promoter fragment (*Gerety et al., 2013*), which drives expression in the lateral line primordium and periderm, among other tissues. This construct was cloned between sites for the Tol1 transposon (*Koga et al., 2008*) and 20 ng of plasmid DNA was injected with 80 ng of *tol1* mRNA into one-cell stage zebrafish embryos. Founders were screened by fluorescence for high expression in the lateral line primordium.

For generation of chimeric PLLp, embryos were dechorionated at ~2 hpf and placed in embryo media with 100 U/mL penicillin and 0.1 mg/mL streptomycin (Roche). When the embryos had reached high-sphere stage (~3.5-4hpf), they were placed in individual wells made in agarose by a custom-printed mold. An Eppendorf CellTram Vario connected to a glass capillary needle with the tip removed at approximately the diameter of an embryonic cell was used to gently aspirate cells from the donor embryo and place them in the host embryo. After transplantation, embryos were placed in individual chambers of a 48-well plate in embryo medium with Penicillin and Streptomycin and grown overnight at 28°C. Embryos were screened at 24hpf for expression of the donor transgene in the PLLp.

## Time-lapse microscopy, segmentation, and quantification

For time-lapse microscopy, embryos were anesthetized in embryo media containing 600 µM MS-222 (Sigma) and mounted in 1% low melt agarose (NuSieve GTG). For Fgf receptor inhibition experiments, embryos were treated with 20 µM SU5402 (*Sun et al., 1999*)(Tocris) for 6 hr prior to imaging. For skin removal experiments, embryos were embedded in 2% low-melt agarose in Fluorobrite media. Agarose above the PLLp was removed with forceps, and a tungsten needle (Roboz surgical instruments) was used to manually remove the skin above the PLLp. Following skin removal, embryos were dissected from the agarose and re-embedded in 1% low-melt agarose in Fluorobrite for imaging. For Matrigel embedding experiments, Matrigel (Corning) was thawed overnight at 4°C and kept on ice. Embryos were first mounted in agarose for skin removal as described above, dissected from agarose, and re-embedded in 20 mg/mL Matrigel with 600 µM MS-222. At this point, the Matrigel was allowed to solidify for 5 min at room temperature before being covered in Fluorobrite. For primordium height measurements, embryos were mounted in 1% agarose in Fluorobrite. The agrose over the trunk was then removed and confocal z-stacks of primordia were acquired with 0.5 µm spacing. The skin over the primordium was then removed as described above, the embryos left for 2 min and then imaged again with the same settings. Height was quantified by making four equally spaced measurements along the primordium and averaging the measurements.

High-resolution time-lapse microscopy of superficial protrusions was performed on a Zeiss 880 Airyscan confocal microscope using the fast Airyscan mode and processed using the default parameters. All images acquired using Airyscan are noted in the corresponding figure legends. For time-lapse imaging of skin healing, imaging was performed on either a Zeiss 880 in regular (diffraction limited) confocal mode, or on a Leica SP5 confocal microscope. Actin rods were manually counted and quantified in FIJI (*Schindelin et al., 2012*) using sum slice projections of processed Airyscan

image stacks taken at 5 s intervals at the PLLp-skin boundary. All rods quantified were within 4.5 μm of the basal surface of the skin (for superficial protrusions) or within 4.5 μm of the basal surface of the PLLp (for basal protrusions). Bleb fluorescence intensity quantification was performed by measuring the intensity of either LifeAct-Citrine or Lyn-mScarlet along the edge of the bleb in single confocal slices. Intensities were normalized to that of the first image in the blebbing sequence. Retrograde actin flow was quantified by making kymographs along the lamellipodia parallel to the direction of flow and speed of flow was quantified by measuring the angle of diagonal lines in the kymographs, representing flow of intensity inhomogeneities in the LifeAct signal (see *Figure 6—figure supplement 1B* for an example).

Segmentation was performed using the Allen Institute for Cell Science Segmenter Python library (*Chen et al., 2018*). Briefly, seed positions were generated using a nuclear image and these seeds were used to initialize a watershed segmentation algorithm which generated individual segmented cells. These segmented objects were converted to a list of 3D vertices using the lewiner marching cubes algorithm (*Lewiner et al., 2003*) in scikit-image (*van der Walt et al., 2014*). Pointclouds derived from these vertices were meshed in Meshlab 2016.12 (*Cignoni et al., 2008*) using the screened Poisson surface reconstruction algorithm (*Kazhdan and Hoppe, 2013* ) followed by simplification by quadratic edge collapse decimation. Rendering was performed in Blender 2.8 (*Blender Online Community, 2018*) using the Eevee (*Figure 2—figure supplement 1*) and Cycles (*Figure 2*, *Figure 2—figure supplement 2A*, *Figure 2—video 1*) rendering engines.

For long-term timelapse, *Tg(clbnb:lyn-egfp);TgBAC(cxcr4b:h2a-mcherry)* double transgenic embryos at 32hpf were embedded in 1% agarose and imaged on a custom DiSPIM (*Wu et al., 2013*) using a 40X objective after removal of overlying agarose. Image registration and joint deconvolution was performed using a recently improved pipeline that offers greatly increased processing speed (*Guo et al., 2020*). Registration was performed using the methods in Guo et al, and for deconvolution we used 10 iterations of conventional Richardson-Lucy. Image rotation was performed using TransformJ (*Meijering et al., 2001*) Data analysis was performed in Python 3.7 using the SciPy (*Jones et al., 2001*), Pandas (*McKinney, 2010*), and NumPy (*Harris et al., 2020*) libraries. Plots were generated in Python using the Matplotlib (*Hunter, 2007*) and Seaborn (*Waskom et al., 2014*) libraries, and in SAS V9. Number of embryos used for each experiment can be found in *supplementary file 1*. Raw data in CSV format, Jupyter notebooks and SAS reports to recreate all the plots and statistical analyses shown are available at (https://github.com/chitnislabnih/dallenogare2019; *Nogare, 2019* copy archived at swh:1:rev:cf258e888577731586096fd464d34df1bd0b2c1a).

## Acknowledgements

The authors wish to thank Holger Knaut and Alvaro Sagasti for generously sharing transgenic lines, Angela Hvitved for editing the manuscript, and Chongmin Wang for statistical analysis support. This work was supported by the intramural research program of the Eunice Kennedy Shriver National Institute of Child Health and Human Development at the National Institutes of Health (HD001012) and the intramural research program of the National Institute of Biomedical Imaging and Bioengineering at the National Institutes of Health. All work was performed in accordance with animal study protocol 18–013.

## Additional information

### Funding

| Funder | Grant reference number | Author |
|---|---|---|
| Eunice Kennedy Shriver National Institute of Child Health and Human Development | HD001012 | Damian E Dalle Nogare<br>Naveen Natesh<br>Ajay B Chitnis |
| National Institute of Biomedical Imaging and Bioengineering | | Harshad D Vishwasrao<br>Hari Shroff |

The funders had no role in study design, data collection and interpretation, or the decision to submit the work for publication.

## Author contributions
Damian E Dalle Nogare, Conceptualization, Data curation, Software, Formal analysis, Supervision, Validation, Investigation, Visualization, Methodology, Writing - original draft, Project administration, Writing - review and editing; Naveen Natesh, Investigation, Writing - review and editing; Harshad D Vishwasrao, Resources, Software, Writing - review and editing; Hari Shroff, Resources; Ajay B Chitnis, Resources, Supervision, Funding acquisition, Investigation, Project administration, Writing - review and editing

## Author ORCIDs
Damian E Dalle Nogare https://orcid.org/0000-0002-9615-9299
Ajay B Chitnis https://orcid.org/0000-0002-6423-6624

## Ethics
Animal experimentation: All work was performed in accordance with IACUC approved animal study protocol #18-013 of the National Institute of Child Health and Human Development at the National Institutes of Health.

## Decision letter and Author response
Decision letter https://doi.org/10.7554/eLife.58251.sa1
Author response https://doi.org/10.7554/eLife.58251.sa2

# Additional files

## Supplementary files
• Supplementary file 1. Table of number of observations and embryos used for determining protrusion length and directionality in *Figures 3*, *6* and *7*.

• Source data 1. Raw data (csv format) and Jupyter notebooks/SAS reports containing all statistical analysis performed.

• Transparent reporting form

## Data availability
CSV files of raw measurements and notebooks containing all statistical analyses are available at Github: https://github.com/chitnislabnih/dallenogare2019 (copy archived at https://archive.software-heritage.org/swh:1:rev:cf258e888577731586096fd464d34df1bd0b2c1a/).

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
