## [Decision Letter]

**Acceptance summary:**

This beautiful imaging study shows that the posterior lateral line primordium in zebrafish is covered by a sheath of cells on its apical side that extend lamellipodial-like protrusion underneath the skin, very similar to the basal cells that extend lamellipodial-like protrusion on top of the muscle. Removal of the skin changes the protrusions – they become bleb-like and disoriented – and the primordium ceases to migrate directionally. Replacing the skin with Matrigel or sealing the wound partly or completely restore protrusions and migration.

**Decision letter after peer review:**

Thank you for submitting your article "A sheath of motile cells supports collective migration of the Zebrafish posterior lateral line primordium under the skin" for consideration by *eLife*. Your article has been reviewed by three peer reviewers, and the evaluation has been overseen by a Reviewing Editor and Didier Stainier as the Senior Editor. The reviewers have opted to remain anonymous.

The reviewers have discussed the reviews with one another and the Reviewing Editor has drafted this decision to help you prepare a revised submission.

Summary:

This is a beautiful imaging study that identifies a previous overlooked cell population in the posterior lateral line primordium in zebrafish – a model for collective cell migration. This cell population is sandwiched between the two-layered skin of the zebrafish embryo and the more basally located main cell population of the primordium. Similarly, to the basal cell population which extends protrusion on the basal side of the primordium, this new cell population also extends protrusions to the apical side between the skin and the primordium. Beautiful microsurgery experiments combined with high resolution imaging shows that skin removal causes the cells of the primordium to stall and change their protrusion type from lamellipodia-like to bleb-like, suggesting that the skin is required to maintain the correct chemical milieu and/or physical pressure to support the migration of the primordium. The identification of this cell population and the requirement of the skin for migration are important contribution to our understanding of tissue migration.

Overall, this manuscript is well written and compelling. We have the following suggestions to improve the manuscript.

Essential revisions:

1) Confinement of the primordium. The reviewers agree that the height of the primordium cells with and without skin should be measured to see if the primordium cells are higher along the medial-lateral axis after skin removal. If the cells are higher, this would be in agreement with the mechanical confinement idea. However, all reviewers feel that the skin removal experiments do not show that confinement is essential for primordium migration and this claim should be softened and other possibilities should be discussed (changes in the environment, wounding response etc).

2) Sheath cells.

a) The reviewers agree that it is unclear (and unlikely) that the sheath cells are a separate cell population. The authors should rename this cell population using a more descriptive term such as apical cells.

b) The claim that the sheath cells are supporting collective migration should be toned down since the authors do not direct evidence for this.

c. The claim that the sheath cells are mesenchymal should be softened since they are labeled by a epithelial promoter.

3) Discussion. The Discussion should be more focused and concise as laid out in the individual reviewer's comments.

Reviewer #1:

The lateral line primordium (LLP) of zebrafish has long been a premier system to study the molecular and cellular mechanisms that underly collective cell migration. In this manuscript, Dalle Nogare et al. identify a new cell population that is essential for the movement of the LLP. Specifically, they describe a superficial sheet of cells that lies between the main primordium and the skin. These cells extend polarized protrusions in the direction of tissue movement and appear to work in concert with the protrusions made by deeper cells to allow the LLP to generate tractions forces against both the overlying skin and the underlying somites simultaneously. They further show that, like the deeper cells, the superficial cells require FGF signaling to orient these protrusions. They also perform a clever set of experiments in which they remove sections of the skin and then allow it to heal back over the LLP to show a strong correlation between the presence of the skin, the protrusive behavior of both the deep and superficial cells, and the migratory ability of the LLP as a whole.

Overall, I think this paper is a strong candidate for publication in *eLife*. Although the study is largely descriptive in nature, the discovery of this new cell population is important, as it fundamentally changes the way we think about how motive forces are generated in this collective. Moreover, the skin healing experiments are elegant and quite revealing of the underlying biology.

1) I am unconvinced that the newly described cell population is mesenchymal, as proposed by the authors. Since a claudin promoter is used to label the cells, it seems more likely that they are epithelial.

2) Results – The authors point out that the rate of retrograde actin flow correlates with traction forces in migrating cells. Please provide some context for the flow rate measured in this paper. How does it compare to other cell types and how can this information help us better understand the underlying biology?

3) Figure 6C – Wild-type control data should be added to this figure.

Reviewer #2:

This insightful manuscript shows that the zebrafish posterior lateral line primordium (PLLp) contains protruding cells around its entire circumference, including on the overlying skin. These cells are structurally distinct from mesenchymal leading cells and epithelialized neuromast-forming cells. Technically impressive skin removal experiments with high-resolution imaging demonstrate that the skin has a previously undescribed role in PLLp protrusion formation and migration, demonstrating that the cluster of migrating cells walks on the "ceiling" as well as on the "floor". These striking results will certainly have an impact on the field. However, we feel that some textual revision is necessary for the conclusions to reflect the data presented. In particular, these experiments do not definitively show that the skin's role depends on confinement. We recommend concluding further explanation that the skin's role may be mediated by confinement, adhesion, tissue integrity signaling, or a combination thereof (see comment #1). Our other suggestions could strengthen the paper but are not required.

1) The skin removal experiments require a more nuanced interpretation of the variety of mechanism(s) by which skin may contribute to PLLp migration. We ask that the authors address the following points in the Discussion and Abstract:

a) The term "confinement" needs to be used more precisely. In each condition of the skin removal experiments, the primordium is in contact with a different material with unique mechanical and chemical properties. We caution the authors from referring to embryos embedded in agarose as "unconfined", as agarose often acts as a form of confinement for cultured cells. Perhaps the PLLp experiences weaker forces or geometric constraints while embedded in agarose versus contained within the skin, but the authors should more clearly acknowledge these unknown mechanical parameters that are inherent to the experiment (Results and Discussion).

The authors could additionally choose to strengthen the mechanical comparison by measuring the height of the PLLp under each condition--one would expect that taller cells are less confined. This measurement should be possible by further analysis of image data that the authors have already collected.

b) The possible role of adhesions should be elevated. We find this model interesting, and the slow migration under Matrigel does not technically prove that adhesion is insufficient for PLLp migration, as the authors correctly note in the Discussion (eighth paragraph). It is possible that growth factors in Matrigel alter chemotaxis, counteracting the pro-migratory effect of lamellipodial formation.

If experimental follow-ups were feasible, we would propose observing transplanted basal cells under Matrigel confinement, in order to determine whether superficial lamellipodia rescue basal cryptic lamellipodia with this perturbation.

c) Tissue integrity should be discussed as a potential contributor to the skin effect. There are multiple signalling pathways triggered by breaking the skin barrier, including calcium signaling, reactive oxygen species, purinergic signaling, and cell swelling (c.f. Enyedi and Niethammer (2015). Trends in Cell Biology 25(7): 398-407), any of which could potentially also interfere with PLLp migration. Because protrusions resume on cells near the front of the primordium as soon as they come into contact with the skin, but directional movement does not actually start up again until the skin is fully healed, it seems plausible that the cells are detecting both the adhesive contact and also some other property of tissue integrity.

The experiments shown in Figure 4 could shed light on this hypothesis if the authors were to additionally measure cell speed after the skin has entirely closed (and presumably skin integrity restored) and compare that with speeds before/after cell contact with skin (Figure 4G).

2) When defining the sheath of motile cells, we suggest including a more thorough analysis of the entire PLLp population, rather than just the cells above apical constrictions (Figure 2). Specifically, the authors define the relevant cell population as those which are in direct physical contact with the skin and also whose nuclei are above a rosette. Why are both criteria necessary? Another perspective on the functional roles of the two distinct populations of actively motile cells in migration of the primordium might explore more completely the relative contributions of those that are walking on the "floor" versus the "ceiling". For example, what proportion of cells are touching the skin versus the basal substrate? Of these populations, how many cells have apical constrictions? The authors could better contextualize these results by citing any previous ultrastructural work on the PLLp, or other work that would shed light on the arrangement of cells within and around the PLLp. Also, the authors should give more prominence to their striking observation that the skin was removed without the PLLp 95% of the time. This result suggests functional importance of differential adhesion among the different cell types. Analogous findings on the significance of differential cell adhesion have been reported in other developmental contexts, for example Tsai et al., bioRxiv 803635.

3) While there are a lot of interesting issues raised in the Discussion, we found it to be unfocused and confusing. We suggest focusing on two key trends in order to consolidate the studies in the Discussion and provide a more cogent summary of how their new results fit in with previous work. First, in both this and previous studies, an environment with more restrictive geometry and/or more adhesivity promotes cell polarity and persistence (Ruprecht et al., 2015; Liu et al., 2015; Bergert et al., 2012; Ramalingam et al., 2015 Nat Comm 6:8496; Logue et al., 2015 *eLife* 4:e08314; Graziano et al., 2019 PLoS Biol 17(10): e3000457). The second key trend is that protrusion type is determined by the interplay of adhesion, confinement, contractility, and polymerization. In agreement with this study, Ruprecht et al., Liu et al., Bergert et al., show that low adhesivity promotes blebs over lamellipodia. However, two *Dictyostelium* studies show a protrusion type trend that is less consistent with this study, because Dicty produces more blebs under greater confinement (Ibo et al., 2016; Srivastava et al., 2020 PNAS 117 (5) 2506-2512). The authors note this, but do not explicitly address differential adhesion. It might be helpful for the authors to express their opinions about the limitations of comparing different cell types, particularly in the context of the ~billion year separation between cellular slime molds and vertebrate animals. We are not convinced by the statement that the results of Bergert et al., 2012, are "less consistent" with the observations in the primordium than the other works cited. In particular, the result that the primordium produced more lamellipodia on Matrigel than on agarose is consistent with the Bergert et al. experiment discussed by the authors, which shows that Walker 256 cells produce more lamellipodia on adhesive regions as compared to non-adhesive regions (Figure 5 in Berget et al.).

In the present version of the manuscript, the Abstract emphasizes the importance of confinement, but the results are also consistent with a more nuanced interpretation where confinement acts along with adhesion and signaling in determining cell behavior.

Reviewer #3:

The manuscript by Dalle Nogare and colleagues presents two interesting ideas. The first is that a collectively migrating tissue, the zebrafish PLLp, uses a 'circumferential' or 'chimneying' motility mechanism, where traction is generated on underlying muscles and overlying skin. This novel idea is supported with cutting-edge, high resolution live imaging data. The second is that the 'confinement' of the tissue between muscles and skin regulates the type of protrusions that these motile cells generate in vivo, an idea supported with technically-impressive skin removal experiments. The study is well written and presented and the findings are potentially of sufficient importance and novelty to justify publication in *eLife*. However, neither conclusion is sufficiently supported by the data presented. If the authors can strengthen EITHER conclusion with additional experimental support, they will make an important contribution to the field.

1) Stunning time-lapse data reveal that cells on the apical surface of the PLLp display protrusions that are consistent with the idea that they are migrating on the overlying skin layer. However, imaging data alone are insufficient to conclude that this motile 'sheath' of cells supports the collective migration of the PLLp, as stated in the title. The only experimental test of this idea – removal of the overlying skin – results in a very strong defect in protrusions on both sides of the primordium, and thus does not reveal a specific requirement for the newly discovered apical motility.

Recommendation 1: Provide more direct evidence for the proposal that apical cells support PLLp migration by interacting with the overlying skin. Ideally, the authors could use their genetic mosaic technique to target these cells directly (rather than disrupting the skin). Alternatively, it may be possible to apply correlative approaches on pre-existing imaging data to further support the suggestion that the dynamic behavior of sheath cells has an influence on tissue migration. For example, do changes in sheath cell behavior, perhaps when they move from one skin cell to the next, have any effect on the migration of the PLLp (e.g. shearing) as might be expected from a chimneying model?

Comment: It may be argued that the imaging data, which clearly show the motile behavior of apical sheath cells on the skin, are sufficient to demonstrate the existence of circumferential motility mechanism. In that case, the authors should tone down their claim that these cells 'support' collective migration.

2) The other major claim is that the study 'highlights the broad role confinement plays in the formation of polarized lamellipodia'. While 'confinement' is not clearly defined, the discussion centres around proposed roles for mechanical confinement in regulating protrusion types that have emerged from in vitro model systems. Here, the support for a role for mechanical confinement comes from skin removal experiments. The issue is that this experiment does more than reduce mechanical confinement, it also severely disrupts what is normally a closed and controlled chemical environment. It's hardly surprising that cells that normally migrate in the presence of patterned diffusible cues and organized matrices suddenly arrest when their closed and controlled environment is forced open.

Recommendation 2: In order to support the proposed role for mechanical confinement in the formation of lamellipodia the authors should rule out that the observed changes aren't the result of chemical effects such as osmotic changes. Do cells of the rosette structures – that don't normally show apical protrusions – start to bleb apically after skin removal? Alternatively, more direct evidence for the proposed role for mechanical confinement in regulating protrusion type could come from experiments that emulate in vitro studies where 'purely mechanical' changes can be applied (by placing cells under agarose pads, for example). The finding that the authors can replace some aspects of the skin with Matrigel is a promising step in this direction. Did the authors consider modulating the elastic modulus of the Matrigel via different concentrations to address the impact of altered mechanics more specifically? The choice of Matrigel concentration is not explained in the Materials and methods section. Better still, the authors could consider replacing Matrigel with agarose to alter mechanical confinement in a way analogous to the under agarose in vitro experiments.

---

## [Author Response]

Dear Editors and Reviewers,

Thank you for your thorough consideration and review of our paper. To address the concerns of the reviewers (listed under the “essential revisions”), we have made the following modifications. In some cases, we have performed additional experiments, however due to the restricted work schedule we are under because of COVID-19, we could not experimentally address all of the reviewers' criticisms. However, we hope the revised manuscript satisfies the majority of the reviewers’ concerns.

Additional experiments and the role of superficial cells In addition to the experiment described above to examine confinement of the PLLp, we have also performed the experiment suggested by reviewer #2 looking at basal protrusions in skinned but matrigel embedded embryos. We report that, unlike superficial protrusions, the directionality of basal protrusions is rescued when embryos are embedded in matrigel. This surprising result further argues against a purely mechanical confinement mechanism (as recovery of polarization is not observed after agarose embedding), and so we have, as discussed above, softened the relevant conclusions. However, despite recovery of polarized basal protrusions, directed primordium migration is not recovered in matrigel embedded embryos. We feel that this failure to migrate, when basal protrusions are polarized but superficial protrusions are not, provides stronger support for the necessity of polarized superficial protrusions in migration. We have discussed this experiment in the relevant portions of the results and discussion. In addition, we would like to address the following comments by the reviewers directly .

Reviewer #1

Actin flow rates: We have added some discussion of actin flow rates in a variety of other contexts to better situate our measurements. As we note, measurement of Actin retrograde flow rates in vivo are very rare, and often significantly different from those observed in in vitro studies, and so we refrain from interpreting too much from our measurement, other than situating it in context of other measured rates (both in vivo and in vitro).

Reviewer #2

As discussed above, we have made significant changes to the way we discuss “confinement” in the manuscript, including softening conclusions about the role of confinement, given the new results reported in Figure 7. We agree with the reviewer that we could have done a more thorough job in describing the nature of this population. To this end, we have added a section in the results describing external contacts of each cell population, which is further explained and quantified in a new Figure S2. In addition, we have provided two examples of the morphology of leading, protoneuromast, and superficial cells in a new Figure 2 – figure supplement 1. We feel that these new figures and results text will do a better job giving context to the reader. We have rewritten parts of the results and discussion to increase clarity, and included a supplementary table showing the number of embryos and movies used for each quantification. In addition, we have added a new supplementary figure with examples of each type of quantification used (actin rods in a variety of contexts, retrograde flow kymographs). We have also added a discussion of these measurements to the main text.

Reviewer #3

We have made all of the modifications in the text and figures suggested by this reviewer. We hope that the additional experiments described help to alleviate their major concerns, however we regret that we were unable to perform some of the experiments suggested. We have, in these cases, toned down the relevant conclusions. Some experiments (for example modulating the elastic modulus of matrigel) we plan to perform in upcoming studies. We agree with the reviewer that figure S1 showing a track of 5 nuclei is underwhelming. We have replaced Figure S1 with a more robust figure and movie (Figure 2 – figure supplement 2, Movie S3). In this figure, we segment, 3D reconstruct and track every nucleus in a primordium over two hours. This data is used to construct a new figure showing the averaged nuclear position of all nuclei in a 120 micron cross section of the PLLp (centered around a neuromast). We have also included a reconstruction movie of all nuclei in the PLLp over this time course, color coded by cell type, and showing the cross section and position heatmaps as movie S3, and the area used to generate the position heatmap is indicated in this movie. We also agree that we do not have complete enough data to make firm statements about the ultimate fate of these cells. Although we strongly suspect that they become interneuromast cells, we have not conclusively demonstrated this, and so have removed this claim from the paper.

Essential revisions:1) Confinement of the primordium. The reviewers agree that the height of the primordium cells with and without skin should be measured to see if the primordium cells are higher along the medial-lateral axis after skin removal. If the cells are higher, this would be in agreement with the mechanical confinement idea.

While we could extract some values from our existing data, we felt the better approach was to perform an additional experiment to quantify this potential change. We have therefore performed additional paired measurements of the height of the PLLp in primordia before and after skin removal. These data are discussed in the Results section, and presented in Figure 4. Removal of the skin results in an almost 20% increase in average primordium height (p = 6e^-7^, paired t-test). Note that for this measurement, we could not robustly measure individual cell height paired between skin intact and skin removed, and so we have devised an average measure of primordium height, measured at 4 equally spaced points along the length of the primordium, as detailed in the Materials and methods.

However, all reviewers feel that the skin removal experiments do not show that confinement is essential for primordium migration and this claim should be softened and other possibilities should be discussed (changes in the environment, wounding response etc).

Despite the results described above suggesting that the primordium is in fact under mechanical confinement by the skin, we agree with the reviewers critique that our experiments do not adequately demonstrate that mechanical confinement, per se, is essential for migration. We have toned down the relevant conclusions and expanded discussion of additional ways confinement by the skin could contribute to migration.

2) Sheath cells.a) The reviewers agree that it is unclear (and unlikely) that the sheath cells are a separate cell population. The authors should rename this cell population using a more descriptive term such as apical cells.

We agree with the reviewers that this population need not represent a separate cell population with a distinct fate. It could, for example, represent the persistence of mesenchymal-like cells that are present in the leading domain. We have rewritten the relevant sections to provide better context for this population, and now refer to these cells as “superficial primordium cells”, rather than “sheath cells”, which inadvertently suggested a distinct cell type. We have, in a small number of places, retained the use of the word “sheath” to describe the overall structure as we feel this is a good intuitive analogy to their position wrapping around the top and sides of the primordium.

b) The claim that the sheath cells are supporting collective migration should be toned down since the authors do not direct evidence for this.c) The claim that the sheath cells are mesenchymal should be softened since they are labeled by a epithelial promoter.

With regards to the mesenchymal/epithelial nature of these cells, we agree and have changed the text to reflect the ambiguity about the nature of these cells, and included a caveat that they are labeled by an epithelial promoter. We also note that the leading cells of the primordium, commonly referred to in the literature of “mesenchymal” or “mesenchymal-like” are also labelled by the cldnB promoter. We have changed text referring to these types of cells as “mesenchymal-like” rather than “mesenchymal” and discuss them now in the context of polarized epithelial cells incorporated into rosettes as opposed to cells in the leading domain and superficial cells that are not incorporated into a neuromast.

3) Discussion. The Discussion should be more focused and concise as laid out in the individual reviewer's comments.

We have significantly rewritten the discussion and hope that is more precise and clearer than the initially submitted manuscript. In addition, to address the reviewers concerns about the legibility of the figures, we have made several changes to font sizes etc to enhance readability. To this end, we have split Figure 5 (which had panels A-Zz) into a new Figure 5 and Figure 6 to allow for larger panels and a re-designed figure.

Reviewer #1:Overall, I think this paper is a strong candidate for publication in eLife. Although the study is largely descriptive in nature, the discovery of this new cell population is important, as it fundamentally changes the way we think about how motive forces are generated in this collective. Moreover, the skin healing experiments are elegant and quite revealing of the underlying biology.1) I am unconvinced that the newly described cell population is mesenchymal, as proposed by the authors. Since a claudin promoter is used to label the cells, it seems more likely that they are epithelial.

We have changed the use of “mesenchymal” in the paper, and have added a section noting that these cells are labelled with the claudinB promoter

2) Results – The authors point out that the rate of retrograde actin flow correlates with traction forces in migrating cells. Please provide some context for the flow rate measured in this paper. How does it compare to other cell types and how can this information help us better understand the underlying biology?

We have added some discussion of other flow rates measured in a variety of contexts to this part of the Results section and attempted to do a better job contextualizing this flow rate

3) Figure 6C – Wild-type control data should be added to this figure.

We have added wild-type (skin intact) data, from embryos embedded both in agarose and matrigel, to this figure.

Reviewer #2:[…] We recommend concluding further explanation that the skin's role may be mediated by confinement, adhesion, tissue integrity signaling, or a combination thereof (see comment #1). Our other suggestions could strengthen the paper but are not required.1) The skin removal experiments require a more nuanced interpretation of the variety of mechanism(s) by which skin may contribute to PLLp migration. We ask that the authors address the following points in the Discussion and Abstract:a) The term "confinement" needs to be used more precisely. In each condition of the skin removal experiments, the primordium is in contact with a different material with unique mechanical and chemical properties. We caution the authors from referring to embryos embedded in agarose as "unconfined", as agarose often acts as a form of confinement for cultured cells. Perhaps the PLLp experiences weaker forces or geometric constraints while embedded in agarose versus contained within the skin, but the authors should more clearly acknowledge these unknown mechanical parameters that are inherent to the experiment (Results and Discussion).

We agree with the reviewer and have modified the text to more precisely use the word “confinement”.

The authors could additionally choose to strengthen the mechanical comparison by measuring the height of the PLLp under each condition--one would expect that taller cells are less confined. This measurement should be possible by further analysis of image data that the authors have already collected.

We have performed experiments to measure this, as discussed earlier, and now show that the primordium is ~15-20% taller on average when the skin is removed than when it is in intact. As we feel these data add more to the narrative than the current Figure 4H (which showed data to support Figure 4G but didn’t add anything new), we have made these height measurements Figure 4H and removed the old Figure 4H.

b) The possible role of adhesions should be elevated. We find this model interesting, and the slow migration under Matrigel does not technically prove that adhesion is insufficient for PLLp migration, as the authors correctly note in the Discussion (eighth paragraph). It is possible that growth factors in Matrigel alter chemotaxis, counteracting the pro-migratory effect of lamellipodial formation.

We have added discussion of the potential role of adhesions and other potential matrigel-related affects to the Discussion.

If experimental follow-ups were feasible, we would propose observing transplanted basal cells under Matrigel confinement, in order to determine whether superficial lamellipodia rescue basal cryptic lamellipodia with this perturbation.

We have performed these experiments, and now show that basal protrusions are in fact recovered after matrigel embedding. Surprisingly, we also show that these basal protrusions (unlike the superficial protrusions) are in fact polarized in matrigel. This may, in part, explain the limited recovery of migratory ability in these primordia when compared to primordia embedded in agarose. The reason for this asymmetric recovery is unclear.

c) Tissue integrity should be discussed as a potential contributor to the skin effect. […] The experiments shown in Figure 4 could shed light on this hypothesis if the authors were to additionally measure cell speed after the skin has entirely closed (and presumably skin integrity restored) and compare that with speeds before/after cell contact with skin (Figure 4G).2) When defining the sheath of motile cells, we suggest including a more thorough analysis of the entire PLLp population, rather than just the cells above apical constrictions (Figure 2). […] Analogous findings on the significance of differential cell adhesion have been reported in other developmental contexts, for example Tsai et al., bioRxiv 803635.

We have now performed a more thorough analysis of the nature of this cell population from segmented data, and answered several of these questions. We have redesigned Figure 2—figure supplement 1 to include some of these data on cell morphology and external contacts.

3) While there are a lot of interesting issues raised in the Discussion, we found it to be unfocused and confusing. […] In particular, the result that the primordium produced more lamellipodia on Matrigel than on agarose is consistent with the Bergert et al. experiment discussed by the authors, which shows that Walker 256 cells produce more lamellipodia on adhesive regions as compared to non-adhesive regions (Figure 5 in Berget et al.).

We have rewritten the Discussion to attempt to address these points.

In the present version of the manuscript, the Abstract emphasizes the importance of confinement, but the results are also consistent with a more nuanced interpretation where confinement acts along with adhesion and signaling in determining cell behavior.

We agree with the reviewer, and have modified the Results and Discussion to better discuss other interpretations consistent with the data.

Reviewer #3:1) Stunning time-lapse data reveal that cells on the apical surface of the PLLp display protrusions that are consistent with the idea that they are migrating on the overlying skin layer. However, imaging data alone are insufficient to conclude that this motile 'sheath' of cells supports the collective migration of the PLLp, as stated in the title. The only experimental test of this idea – removal of the overlying skin – results in a very strong defect in protrusions on both sides of the primordium, and thus does not reveal a specific requirement for the newly discovered apical motility.Recommendation 1: Provide more direct evidence for the proposal that apical cells support PLLp migration by interacting with the overlying skin. Ideally, the authors could use their genetic mosaic technique to target these cells directly (rather than disrupting the skin). Alternatively, it may be possible to apply correlative approaches on pre-existing imaging data to further support the suggestion that the dynamic behavior of sheath cells has an influence on tissue migration. For example, do changes in sheath cell behavior, perhaps when they move from one skin cell to the next, have any effect on the migration of the PLLp (e.g. shearing) as might be expected from a chimneying model?Comment: It may be argued that the imaging data, which clearly show the motile behavior of apical sheath cells on the skin, are sufficient to demonstrate the existence of circumferential motility mechanism. In that case, the authors should tone down their claim that these cells 'support' collective migration.

We have tried experiments similar to the ones suggested by the reviewer, but have been unable to target these cells with the precision necessary for these experiments, without also damaging surrounding tissues. However, we also now demonstrate that when re-embedded in matrigel basal, but not superficial, protrusion directionality is restored, while migration is still profoundly affected. We argue that this supports our conclusion that polarized superficial protrusive activity is necessary for migration. However, we recognize the limitations of the evidence provided and have we have toned down our claim that these cells support collective migration and added some necessary caveats.

2) The other major claim is that the study 'highlights the broad role confinement plays in the formation of polarized lamellipodia'. While 'confinement' is not clearly defined, the discussion centres around proposed roles for mechanical confinement in regulating protrusion types that have emerged from in vitro model systems. Here, the support for a role for mechanical confinement comes from skin removal experiments. The issue is that this experiment does more than reduce mechanical confinement, it also severely disrupts what is normally a closed and controlled chemical environment. It's hardly surprising that cells that normally migrate in the presence of patterned diffusible cues and organized matrices suddenly arrest when their closed and controlled environment is forced open.Recommendation 2: In order to support the proposed role for mechanical confinement in the formation of lamellipodia the authors should rule out that the observed changes aren't the result of chemical effects such as osmotic changes. […] Better still, the authors could consider replacing Matrigel with agarose to alter mechanical confinement in a way analogous to the under agarose in vitro experiments.

We have considered these experiments, but felt that a more systematic exploration of the mechanical properties of artificial substrates would be better described in a separate manuscript that focuses on these parameters. We have added a sentence describing the rationale for using 20mg/mL matrigel (this was the undiluted concentration, and we reasoned that we would like to provide as stiff a substrate as possible for primordium migration) in the appropriate place in the Results.